# Learning Linear Programs from Optimal Decisions

**Yingcong Tan**[1], **Daria Terekhov**[1], **Andrew Delong**[2]
[1]Mechanical, Industrial and Aerospace Engineering
[2]Computer Science and Software Engineering
Gina Cody School of Engineering and Computer Science
Concordia University, Montréal, Canada
{yingcong.tan, daria.terekhov, andrew.delong}@concordia.ca

## Abstract

We propose a flexible gradient-based framework for learning linear programs from optimal decisions. Linear programs are often specified by hand, using prior knowledge of relevant costs and constraints. In some applications, linear programs must instead be learned from observations of optimal decisions. Learning from optimal decisions is a particularly challenging bilevel problem, and much of the related *inverse optimization* literature is dedicated to special cases. We tackle the general problem, learning all parameters jointly while allowing flexible parametrizations of costs, constraints, and loss functions. We also address challenges specific to learning linear programs, such as empty feasible regions and non-unique optimal decisions. Experiments show that our method successfully learns synthetic linear programs and minimum-cost multi-commodity flow instances for which previous methods are not directly applicable. We also provide a fast batch-mode PyTorch implementation of the homogeneous interior point algorithm, which supports gradients by implicit differentiation or backpropagation.

## 1   Introduction

In linear programming, the goal is to make an optimal decision given a linear objective and subject to linear constraints. Traditionally, a linear program is designed using knowledge of relevant costs and constraints. More recently, methodologies that are data-driven have emerged.

Inverse optimization (IO) [Burton and Toint, 1992, Troutt, 1995, Ahuja and Orlin, 2001], in contrast, learns linear programs from observations of optimal decisions rather than of the costs or constraints themselves. The IO approach is particularly important when observations come from optimizing agents (e.g., experts [Chan et al., 2014, Bärmann et al., 2017] or customers [Dong et al., 2018]) who make near-optimal decisions with respect to their internal (unobserved) optimization models.

From a machine learning perspective, the IO setup is as follows: we are given feature vectors $\{\mathbf{u}_1, \mathbf{u}_2, \ldots, \mathbf{u}_N\}$ representing conditions (e.g., time, prices, weather) and we observe the corresponding decision targets $\{\mathbf{x}_1^{\mathrm{obs}}, \mathbf{x}_2^{\mathrm{obs}}, \ldots, \mathbf{x}_N^{\mathrm{obs}}\}$ (e.g., quantities, actions) determined by an unknown optimization process, which in our case is assumed linear. We view IO as the problem of inferring a constrained optimization model that gives identical (or equivalent) decisions, and which generalizes to novel conditions $\mathbf{u}$. The family of candidate models is assumed parametrized by some vector $\mathbf{w}$.

Learning a constrained optimizer that makes the observations both feasible *and* optimal poses multiple challenges that have not been explicitly addressed. For instance, parameter setting $\mathbf{w}_1$ in Figure 1 makes the observed decision $\mathbf{x}_1^{\mathrm{obs}}$ optimal but not feasible, $\mathbf{w}_2$ produces exactly the opposite result, and some $\mathbf{w}$ values (black-hatched region in Figure 1) are not even admissible because they will result in empty feasible regions. Finding a parameter such as $\mathbf{w}_3$ that is consistent with the observations can be difficult. We formulate the learning problem in a novel way, and tackle it with gradient-based methods despite the inherent bilevel nature of learning. Using gradients from backpropagation or

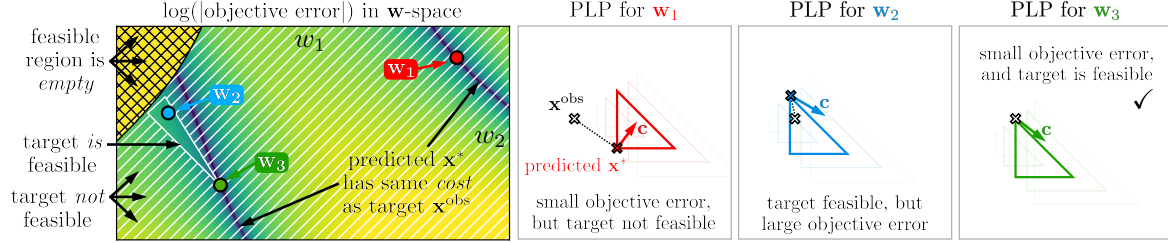

Figure 1: A depiction of our constrained learning formulation. We learn a parametric linear program (PLP), here parametrized by a feature $u$ and weights $\mathbf{w} = (w_1, w_2)$ and using a single training observation $(u_1, \mathbf{x}_1^{\text{obs}})$. The PLP corresponding to three parameter settings $\mathbf{w}_1, \mathbf{w}_2, \mathbf{w}_3$ are shown, with the cost vector and feasible region corresponding to $u_1$ emphasized. The goal of learning is to find solutions such as $\mathbf{w}^* = \mathbf{w}_3$. (See Appendix for the specific PLP used in this example.)

implicit differentiation, we successfully learn linear program instances of various sizes as well as learning the costs and right-hand coefficients of a minimum-cost multi-commodity flow problem.

**Parametric Linear Programs**  In a linear program (LP), the values of decision variables $\mathbf{x} \in \mathbb{R}^D$ must be determined, whereas the cost coefficients $\mathbf{c} \in \mathbb{R}^D$, inequality constraint coefficients $\mathbf{A} \in \mathbb{R}^{M_1 \times D}$, $\mathbf{b} \in \mathbb{R}^{M_1}$, and equality constraint coefficients $\mathbf{G} \in \mathbb{R}^{M_2 \times D}$, $\mathbf{h} \in \mathbb{R}^{M_2}$ are all treated as constants. In a *parametric linear program* (PLP), the coefficients (and therefore the optimal decisions) may depend on features $\mathbf{u}$. In order to infer a PLP from data, one may define a suitable hypothesis space parametrized by $\mathbf{w}$. We refer to this hypothesis space as the form of our *forward optimization problem* (FOP).

$$
\begin{array}{lll}
\min_{\mathbf{x}} \ \mathbf{c}^T\mathbf{x} & \min_{\mathbf{x}} \ \mathbf{c}(\mathbf{u})^T\mathbf{x} & \min_{\mathbf{x}} \ \mathbf{c}(\mathbf{u}, \mathbf{w})^T\mathbf{x} \\
\quad \text{s.t.} \ \mathbf{A}\mathbf{x} \le \mathbf{b} \quad \text{(LP)} & \quad \text{s.t.} \ \mathbf{A}(\mathbf{u})\mathbf{x} \le \mathbf{b}(\mathbf{u}) \quad \text{(PLP)} & \quad \text{s.t.} \ \mathbf{A}(\mathbf{u}, \mathbf{w})\mathbf{x} \le \mathbf{b}(\mathbf{u}, \mathbf{w}) \quad \text{(FOP)} \\
\quad \quad \mathbf{G}\mathbf{x} = \mathbf{h} & \quad \quad \mathbf{G}(\mathbf{u})\mathbf{x} = \mathbf{h}(\mathbf{u}) & \quad \quad \mathbf{G}(\mathbf{u}, \mathbf{w})\mathbf{x} = \mathbf{h}(\mathbf{u}, \mathbf{w})
\end{array}
$$

A choice of hypothesis $\mathbf{w}$ in (FOP) identifies a PLP, and a subsequent choice of conditions $\mathbf{u}$ identifies an LP. The LP can then be solved to yield an optimal decision $\mathbf{x}^*$ under the model. These predictions of optimal decisions can be compared to observations at training time, or can be used to anticipate optimal decisions under novel conditions $\mathbf{u}$ at test time.

## 2 Related Work

**Inverse optimization**  IO has focused on developing optimization models for minimally adjusting a prior estimate of $\mathbf{c}$ to make a single feasible observation $\mathbf{x}^{\text{obs}}$ optimal [Ahuja and Orlin, 2001, Heuberger, 2004] or for making $\mathbf{x}^{\text{obs}}$ minimally sub-optimal to (LP) without a prior $\mathbf{c}$ [Chan et al., 2014, 2019]. Recent work [Babier et al., 2020, Shahmoradi and Lee, 2020] develops exact approaches for imputing non-parametric $\mathbf{c}$ given multiple potentially infeasible solutions to (LP), and to finding non-parametric $\mathbf{A}$ and/or $\mathbf{b}$ [Chan and Kaw, 2020, Ghobadi and Mahmoudzadeh, 2020]. In the parametric setting, joint estimation of $\mathbf{A}$ and $\mathbf{c}$ via a maximum likelihood approach was developed by Troutt et al. [2005, 2008] when only $\mathbf{h}$ is a function of $\mathbf{u}$. Saez-Gallego and Morales [2017] jointly learn $\mathbf{c}$ and $\mathbf{b}$ which are affine functions of $\mathbf{u}$. Bärmann et al. [2017, 2020] and Dong et al. [2018] study online versions of inverse linear and convex optimization, respectively, learning a sequence of cost functions where the feasible set for each observation are assumed to be fully-specified. Tan et al. [2019] propose a gradient-based approach for learning costs and constraints of a PLP, inspired by deep learning: they 'unroll' a barrier interior point solver and backpropagate through its steps. For certain loss functions, their formulation is susceptible to the situation depicted in Figure 1 as '$\mathbf{w}_1$'.

In inverse convex optimization, the focus has been on imputing parametric cost functions while assuming that the feasible region is known for each $\mathbf{u}_i$ [Keshavarz et al., 2011, Bertsimas et al., 2015, Aswani et al., 2018, Esfahani et al., 2018], usually under assumptions of a convex set of admissible $\mathbf{u}$, the objective and/or constraints being convex in $\mathbf{u}$, and uniqueness of the optimal solution for every $\mathbf{u}$. Furthermore, since the feasible region is fixed for each $\mathbf{u}$, it is assumed to be non-empty and bounded, unlike for our work. Although our work focuses on linear programming, it is otherwise substantially more general, allowing for learning of all cost and constraint coefficients simultaneously with no convexity assumptions related to $\mathbf{u}$, no restrictions on the existence of multiple optima, and explicit handling of empty or unbounded feasible regions.

**Optimization task-based learning** Kao et al. [2009] introduces the concept of directed regression, where the goal is to fit a linear regression model while minimizing the decision loss, calculated with respect to an unconstrained quadratic optimization model. Donti et al. [2017] use a neural network approach to minimize a task loss which is calculated as a function of the optimal decisions in the context of stochastic programming. Elmachtoub and Grigas [2020] propose the "Smart Predict-then-Optimize" framework in which the goal is to predict the cost coefficients of a linear program with a fixed feasible region given past observations of features and true costs, i.e., given $(\mathbf{u}_i, \mathbf{c}_i)$. Note that knowing $\mathbf{c}_i$ in this case implies we can solve for $\mathbf{x}_i^*$, so our framework can in principle be applied in their setting but not vice versa. Our framework is still amenable to more 'direct' data-driven prior knowledge: if in addition to $(\mathbf{u}_i, \mathbf{x}_i^*)$ we have partial or complete observations of $\mathbf{c}_i$ or of constraint coefficients, regressing to these targets can easily be incorporated into our overall learning objective.

**Structured prediction** In structured output prediction [Taskar et al., 2005, BakIr et al., 2007, Nowozin et al., 2014, Daumé III et al., 2015], each prediction is $\mathbf{x}^* \in \arg\min_{\mathbf{x} \in \mathcal{X}(\mathbf{u})} f(\mathbf{x}, \mathbf{u}, \mathbf{w})$ for an objective $f$ and known output structure $\mathcal{X}(\mathbf{u})$. In our work the structure is also learned, parametrized as $\mathcal{X}(\mathbf{u}, \mathbf{w}) = \{ \mathbf{x} \mid \mathbf{A}(\mathbf{u}, \mathbf{w})\mathbf{x} \le \mathbf{b}(\mathbf{u}, \mathbf{w}), \mathbf{G}(\mathbf{u}, \mathbf{w})\mathbf{x} = \mathbf{h}(\mathbf{u}, \mathbf{w}) \}$, and the objective is linear $f(\mathbf{x}, \mathbf{u}, \mathbf{w}) = \mathbf{c}(\mathbf{u}, \mathbf{w})^T \mathbf{x}$. In structured prediction the loss $\ell$ is typically a function of $\mathbf{x}^*$ and a target $\bar{\mathbf{x}}$, whereas in our setting it is important to consider a parametric loss $\ell(\mathbf{x}^*, \bar{\mathbf{x}}, \mathbf{u}, \mathbf{w})$.

**Differentiating through optimization** Our work involves differentiating through an LP. Bengio [2000] proposed gradient-based tuning of neural network hyperparameters and, in a special case, backpropagating through the Cholesky decomposition computed during training (suggested by Léon Bottou). Stoyanov et al. [2011] proposed backpropagating through a truncated loopy belief propagation procedure. Domke [2012, 2013] proposed automatic differentiation through truncated optimization procedures more generally, and Maclaurin et al. [2015] proposed a similar approach for hyperparameter search. The continuity and differentiability of the optimal solution set of a quadratic program has been extensively studied [Lee et al., 2006]. Amos and Kolter [2017] recently proposed integrating a quadratic optimization layer in a deep neural network, and used implicit differentiation to derive a procedure for computing parameter gradients. As part of our work we specialize their approach, providing an expression for LPs. Even more general is recent work on differentiating through convex cone programs [Agrawal et al., 2019], submodular optimization [Djolonga and Krause, 2017], and arbitrary constrained optimization [Gould et al., 2019]. There are also versatile perturbation-based differentiation techniques [Papandreou and Yuille, 2011, Berthet et al., 2020].

## 3 Methodology

Here we introduce our new bilevel formulation and methodology for learning parametric linear programs. Unlike previous approaches (e.g., Aswani et al. [2018]), we do not transform the problem to a single-level formulation, and so we do not require simplifying assumptions. We propose a technique for tackling our bilevel formulation with gradient-based non-linear programming methods.

### 3.1 Inverse Optimization as PLP Model Fitting

Let $\{(\mathbf{u}_i, \mathbf{x}_i^{\text{obs}})\}_{i=1}^N$ denote the training set. A loss function $\ell(\mathbf{x}^*, \mathbf{x}^{\text{obs}}, \mathbf{u}, \mathbf{w})$ penalizes discrepancy between prediction $\mathbf{x}^*$ and target $\mathbf{x}^{\text{obs}}$ under conditions $\mathbf{u}$ for the PLP hypothesis identified by $\mathbf{w}$. Note that if $\mathbf{x}_i^{\text{obs}}$ is optimal under conditions $\mathbf{u}_i$, then $\mathbf{x}_i^{\text{obs}}$ must also be feasible. We therefore propose the following bilevel formulation of the *inverse linear optimization problem* (ILOP):

$$\underset{\mathbf{w} \in \mathcal{W}}{\text{minimize}} \quad \frac{1}{N} \sum_{i=1}^N \ell(\mathbf{x}_i^*, \mathbf{x}_i^{\text{obs}}, \mathbf{u}_i, \mathbf{w}) + r(\mathbf{w}) \tag{ILOP}$$

$$\text{subject to} \quad \mathbf{A}(\mathbf{u}_i, \mathbf{w})\mathbf{x}_i^{\text{obs}} \le \mathbf{b}(\mathbf{u}_i, \mathbf{w}), \quad \mathbf{G}(\mathbf{u}_i, \mathbf{w})\mathbf{x}_i^{\text{obs}} = \mathbf{h}(\mathbf{u}_i, \mathbf{w}), \quad i = 1, \dots, N \tag{1a}$$

$$\mathbf{x}_i^* \in \arg\min_{\mathbf{x}} \left\{ \mathbf{c}(\mathbf{u}_i, \mathbf{w})^T \mathbf{x} \;\middle|\; \begin{array}{l} \mathbf{A}(\mathbf{u}_i, \mathbf{w})\mathbf{x} \le \mathbf{b}(\mathbf{u}_i, \mathbf{w}) \\ \mathbf{G}(\mathbf{u}_i, \mathbf{w})\mathbf{x} = \mathbf{h}(\mathbf{u}_i, \mathbf{w}) \end{array} \right\}, \quad i = 1, \dots, N \tag{1b}$$

where $r(\mathbf{w})$ denotes an optional regularization term such as $r(\mathbf{w}) = \|\mathbf{w}\|^2$ and $\mathcal{W} \subseteq \mathbb{R}^K$ denotes additional problem-specific prior knowledge, if applicable (similar constraints are standard in the IO literature [Keshavarz et al., 2011, Chan et al., 2019]). The 'inner' problem (1b) generates predictions $\mathbf{x}_i^*$ by solving $N$ independent LPs. The 'outer' problem tries to make these predictions consistent with the targets $\mathbf{x}_i^*$ while also satisfying target feasibility (1a).

Difficulties may arise, in principle and in practice. An inner LP may be infeasible or unbounded for certain $\mathbf{w} \in \mathcal{W}$, making $\ell$ undefined. Even if all $\mathbf{w} \in \mathcal{W}$ produce feasible and bounded LPs, an algorithm for solving (ILOP) may still attempt to query $\mathbf{w} \notin \mathcal{W}$. The outer problem as a whole may be subject to local minima due to non-convex objective and/or constraints, depending on the problem-specific parametrizations. We propose gradient-based techniques for the outer problem (Section 3.2), but $\frac{d\ell}{d\mathbf{w}}$ may not exist or may be non-unique at certain $\mathbf{u}_i$ and $\mathbf{w}$ (Section 3.3).

Nonetheless, we find that tackling this formulation leads to practical algorithms. To the best of our knowledge, our proposed (ILOP) formulation is the most general model of inverse linear parametric programming. The formulation subsumes cases that are non-parametric, or parametric only in $\mathbf{u}$, that have received much interest in the IO literature. It has not been proposed in work focused purely on differentiation, such as that of Amos and Kolter [2017] or of Agrawal et al. [2019].

**Choice of loss function** The IO literature considers *decision error*, which penalizes difference in decision variables, and *objective error*, which penalizes difference in optimal objective value [Babier et al., 2020]. A fundamental issue with decision error, such as *squared decision error* (SDE) $\ell(\mathbf{x}^*, \mathbf{x}^{\mathrm{obs}}) = \frac{1}{2}\|\mathbf{x}_i^* - \mathbf{x}_i^{\mathrm{obs}}\|^2$, is that when $\mathbf{x}^*$ is non-unique the loss is also not unique; this issue was also a motivation for the "Smart Predict-then-Optimize" paper [Elmachtoub and Grigas, 2020]. An objective error, such as *absolute objective error* (AOE) $\ell(\mathbf{x}^*, \mathbf{x}^{\mathrm{obs}}, \mathbf{c}) = |\mathbf{c}^T(\mathbf{x}_i^{\mathrm{obs}} - \mathbf{x}_i^*)|$, is unique even if $\mathbf{x}^*$ is not. We evaluate AOE using imputed cost $\mathbf{c}(\mathbf{u}, \mathbf{w})$ during training; doing so requires at least some prior knowledge $\mathcal{W}$ to avoid trivial cost vectors, as in Keshavarz et al. [2011].

**Target feasibility** Constraints (1a) explicitly enforce target feasibility $\mathbf{A}\mathbf{x}_i^{\mathrm{obs}} \leq \mathbf{b}$, $\mathbf{G}\mathbf{x}_i^{\mathrm{obs}} = \mathbf{h}$ in any learned PLP. The importance of these constraints can be understood through Figure 1, where hypothesis $\mathbf{w}_1$ achieves AOE $= 0$ since $\mathbf{x}^{\mathrm{obs}}$ and $\mathbf{x}^*$ are on the same hyperplane, despite $\mathbf{x}^{\mathrm{obs}}$ being infeasible. Chan et al. [2019] show that if the feasible region is bounded then for any infeasible $\mathbf{x}^{\mathrm{obs}}$ there exists a cost vector achieving AOE $= 0$.

**Unbounded or infeasible subproblems** Despite (1a), an algorithm for solving (ILOP) may query a $\mathbf{w}$ for which an LP in (1b) is itself infeasible or unbounded, in which case a finite $\mathbf{x}^*$ is not defined. We can extend (ILOP) to explicitly account for these special cases (by penalizing a measure of infeasibility [Murty et al., 2000], and penalizing unbounded directions when detected) but in our experiments simply evaluating the (large) loss for an arbitrary $\mathbf{x}^*$ returned by our interior point solver worked nearly as well at avoiding such regions of $\mathcal{W}$, so we opt to keep the formulation simple.

**Noisy observations** Formulation (ILOP) can be extended to handle measurement noise. For example, individually penalized non-negative slack variables can be added to the right-hand sides of (1a) as in a soft-margin SVM [Cortes and Vapnik, 1995]. Alternatively, a norm-penalized group of slack variables can be added to each $\mathbf{x}_i^{\mathrm{obs}}$ on the left-hand side of (1a), softening targets in decision space. We leave investigation of noisy data and model-misspecification as future work.

### 3.2 Learning Linear Programs with Sequential Quadratic Programming

We treat (ILOP) as a *non-linear programming* (NLP) problem, making as few assumptions as possible. We focus on *sequential quadratic programming* (SQP), which aims to solve NLP problems iteratively. Given current iterate $\mathbf{w}^k$, SQP determines a search direction $\boldsymbol{\delta}^k$ and then selects the next iterate $\mathbf{w}^{k+1} = \mathbf{w}^k + \alpha\boldsymbol{\delta}^k$ via line search on $\alpha > 0$. Direction $\boldsymbol{\delta}^k$ is the solution to a quadratic program.

$$\begin{array}{ll} \text{minimize}_{\mathbf{w}} \ f(\mathbf{w}) & \quad \text{minimize}_{\boldsymbol{\delta}} \ \nabla f(\mathbf{w}^k)^T\boldsymbol{\delta} + \boldsymbol{\delta}^T\mathbf{B}^k\boldsymbol{\delta} \\ \text{subject to} \ \mathbf{g}(\mathbf{w}) \leq \mathbf{0} \quad \text{(NLP)} & \quad \text{subject to} \ \nabla\mathbf{g}(\mathbf{w}^k)^T\boldsymbol{\delta} + \mathbf{g}(\mathbf{w}^k) \leq \mathbf{0} \quad \text{(SQP)} \\ \quad\quad\quad\quad \mathbf{h}(\mathbf{w}) = \mathbf{0} & \quad\quad\quad\quad\quad \nabla\mathbf{h}(\mathbf{w}^k)^T\boldsymbol{\delta} + \mathbf{h}(\mathbf{w}^k) = \mathbf{0} \end{array}$$

Each instance of subproblem (SQP) requires evaluating constraints[1] and their gradients at $\mathbf{w}^k$, as well as the gradient of the objective. Matrix $\mathbf{B}^k$ approximates the Hessian of the Lagrange function for (NLP), where $\mathbf{B}^{k+1}$ is typically determined from the gradients by a BFGS-like update. Our experiments use an efficient variant called *sequential least squares programming* (SLSQP) [Schittkowski, 1982, Kraft, 1988] which exploits a stable $LDL$ factorization of $\mathbf{B}$.

The NLP formulation of (ILOP) has $NM_1$ inequality and $NM_2$ equality constraints from (1a):

$$\mathbf{g}(\mathbf{w}) = \left[\mathbf{A}(\mathbf{u}_i, \mathbf{w})\mathbf{x}_i^{\mathrm{obs}} - \mathbf{b}(\mathbf{u}_i, \mathbf{w})\right]_{i=1}^N, \qquad \mathbf{h}(\mathbf{w}) = \left[\mathbf{G}(\mathbf{u}_i, \mathbf{w})\mathbf{x}_i^{\mathrm{obs}} - \mathbf{h}(\mathbf{u}_i, \mathbf{w})\right]_{i=1}^N,$$

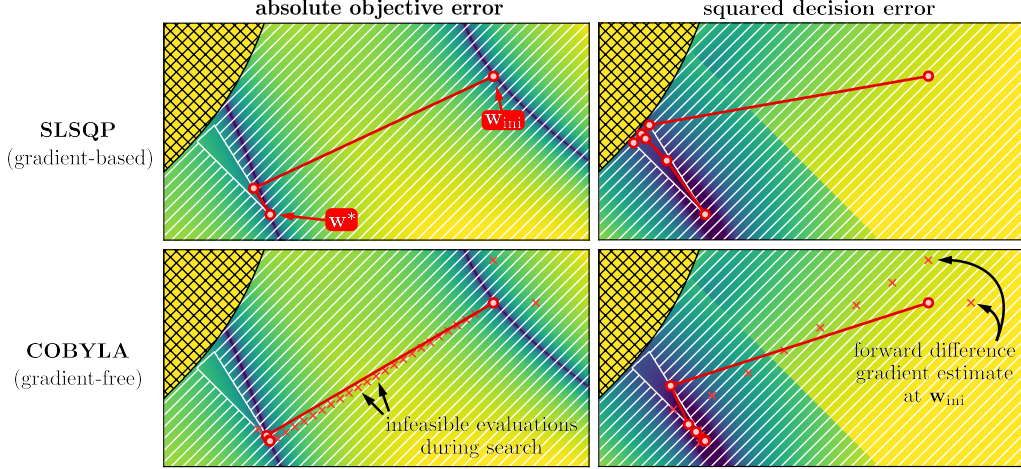

Figure 2: An illustration of how SLSQP and COBYLA solve the learning problem in Figure 1 for the AOE and SDE loss functions. Each algorithm first tries to satisfy the NLP constraints $\mathbf{g}(\mathbf{w}) \leq \mathbf{0}$ (triangle-shaped feasible region in $\mathbf{w}$-space), then makes progress minimizing $f(\mathbf{w})$.

plus any constraints needed to enforce $\mathbf{w} \in \mathcal{W}$. The NLP constraint residuals and their gradients $\nabla \mathbf{g}(\mathbf{w}), \nabla \mathbf{h}(\mathbf{w})$ can be directly evaluated. Evaluating $f(\mathbf{w}) = \frac{1}{N} \sum_{i=1}^{N} \ell(\mathbf{x}_i^*, \mathbf{x}_i^{\mathrm{obs}}, \mathbf{u}_i, \mathbf{w}) + r(\mathbf{w})$ requires solving each LP in (1b). Finally, evaluating $\nabla f(\mathbf{w})$ requires evaluating the vector-Jacobian product term in $\frac{\mathrm{d}\ell}{\mathrm{d}\mathbf{w}} = \frac{\partial \ell}{\partial \mathbf{w}} + \frac{\partial \ell}{\partial \mathbf{x}_i^*} \frac{\partial \mathbf{x}_i^*}{\partial \mathbf{w}}$ for each $i$, which requires differentiating through the LP optimization that produced $\mathbf{x}_i^*$ from $\mathbf{u}_i$ and $\mathbf{w}$. Differentiating through the LP allows us to tackle (ILOP) directly in its bilevel form, using powerful gradient-based NLP algorithms such as SQP as the 'outer' solver. Section 3.3 compares methods for differentiating through an LP optimization.

**Redundant NLP constraints**  When PLP model parameters $\mathbf{w}$ have fixed dimension, the NLP formulation of (ILOP) can involve many redundant constraints, roughly in proportion to $N$. Indeed, if $\mathcal{W} \subseteq \mathbb{R}^K$ and $K < NM_2$, the equality constraints may appear to over-determine $\mathbf{w}$, treating (NLP) as a feasibility problem; but, due to redundancy in (1a), $\mathbf{w}$ is not uniquely determined. The ease or difficulty of removing redundant constraints from (NLP) depends on the domain-specific parametrizations of PLP constraints $\mathbf{A}(\mathbf{u}, \mathbf{w}), \mathbf{b}(\mathbf{u}, \mathbf{w}), \mathbf{G}(\mathbf{u}, \mathbf{w})$, and $\mathbf{h}(\mathbf{u}, \mathbf{w})$. Equality constraints that are affinely-dependent on $\mathbf{w}$ can be eliminated from (NLP) by a pseudoinverse technique, resulting in a lower-dimensional problem; this technique also handles the case where (NLP) is not strictly feasible in $\mathbf{h}(\mathbf{w}) = \mathbf{0}$ (either due to noisy observations or model misspecification) by automatically searching only among $\mathbf{w}$ that exactly minimize the sum of squared residuals $\|\mathbf{h}(\mathbf{w})\|^2$. If equality constraints are polynomially-dependent on $\mathbf{w}$, we can eliminate redundancy by Gröbner basis techniques [Cox et al., 2013] although, unlike the affine case, it may not be possible or beneficial to reparametrize-out the new non-redundant basis constraints from the NLP. Redundant inequality constraints can be either trivial or costly to identify [Telgen, 1983], but are not generally problematic for SQP algorithms. See Appendix for details.

**Benefit over gradient-free methods**  Evaluating $f(\mathbf{w})$ is expensive in our NLP because it requires solving $N$ linear programs. To understand why access to $\nabla f(\mathbf{w})$ is important in this scenario, it helps to contrast SQP with a well-known gradient-free NLP optimizer such as COBYLA [Powell, 1994]. For $K$-dimensional NLP, COBYLA maintains $K + 1$ samples of $f(\mathbf{w}), \mathbf{g}(\mathbf{w}), \mathbf{h}(\mathbf{w})$ and uses them as a finite-difference approximation to $\nabla f(\mathbf{w}^k), \nabla g(\mathbf{w}^k), \nabla h(\mathbf{w}^k)$ where $\mathbf{w}^k$ is the current iterate (best sample). The next iterate $\mathbf{w}^{k+1}$ is computed by optimizing over a trust region centered at $\mathbf{w}^k$. COBYLA recycles past samples to effectively estimate 'coarse' gradients, whereas SQP uses gradients directly. Figure 2 shows SLSQP and COBYLA running on the example from Figure 1.

### 3.3  Computing Loss Function and its Gradients

If, at a particular point $(\mathbf{u}_i, \mathbf{w})$, each corresponding vector-Jacobian product $\frac{\partial \ell}{\partial \mathbf{x}_i^*} \frac{\partial \mathbf{x}_i^*}{\partial \mathbf{w}}$ exists, is unique, and can be computed, then we can construct (SQP) at each step. For convenience, we assume that $(\mathbf{c}, \mathbf{A}, \mathbf{b}, \mathbf{G}, \mathbf{h})$ are expressed in terms of $(\mathbf{u}, \mathbf{w})$ within an automatic differentiation framework such

as PyTorch, so all that remains is to compute Jacobians $(\frac{\partial \ell}{\partial \mathbf{c}}, \frac{\partial \ell}{\partial \mathbf{A}}, \frac{\partial \ell}{\partial \mathbf{b}}, \frac{\partial \ell}{\partial \mathbf{G}}, \frac{\partial \ell}{\partial \mathbf{h}})$ at each $(\mathbf{u}_i, \mathbf{w})$ as an intermediate step at the outset of computing $\frac{d\ell}{d\mathbf{w}}$. We consider four approaches:

*backprop:* backpropagate through the steps of the homogeneous interior point algorithm for LPs,

   *implicit:* the implicit differentiation procedure of Amos and Kolter [2017] specialized to LPs,

      *direct:* evaluate gradients directly, in closed form (for objective error only), and

         *cvx:* use a *cvxpylayer* [Agrawal et al., 2019] for LP solve and for implicit differentiation.

To implement the first three approaches, we developed a batch PyTorch version of the homogeneous interior point algorithm [Andersen and Andersen, 2000, Xu et al., 1996]; this algorithm was originally developed for the MOSEK optimization suite and is currently the default linear programming solver in SciPy [Virtanen et al., 2020]. Our backprop implementation is also efficient, for example re-using the $LU$ decompositionfrom each Newton step.

For implicit differentiation we follow Amos and Kolter [2017] by forming the system of linear equations that result from differentiating the KKT conditions and then inverting that system to compute the needed vector-Jacobian products. For LPs this system can be poorly conditioned, especially at strict tolerances on the LP solver, but in practice it provides useful gradients. The *cvx* approach is similar but is implemented by a *cvxpylayer*, which in turn relies on a fast conic solver [O'Donoghue et al., 2016] for the forward problem and an implicit differentiation procedure similar to the work of Amos and Kolter [2017] for the gradients.

For direct gradients (in the case of objective error), we use Theorem 1.

**Theorem 1.** *Let* $\mathbf{x}^* \in \mathbb{R}^D$ *be an optimal solution to* (LP) *and let* $\boldsymbol{\lambda}^* \in \mathbb{R}^{M_1}_{\leq 0}, \boldsymbol{\nu}^* \in \mathbb{R}^{M_2}$ *be an optimal solution to the associated dual linear program. If* $\mathbf{x}^*$ *is non-degenerate then the objective error* $z = \mathbf{c}^T(\mathbf{x}^{\text{obs}} - \mathbf{x}^*)$ *is differentiable and the total derivatives[2] are*

$$\frac{\partial z}{\partial \mathbf{c}} = \left(\mathbf{x}^{\text{obs}} - \mathbf{x}^*\right)^T \qquad \frac{\partial z}{\partial \mathbf{A}} = \boldsymbol{\lambda}^* \mathbf{x}^{*T} \qquad \frac{\partial z}{\partial \mathbf{b}} = -\boldsymbol{\lambda}^{*T} \qquad \frac{\partial z}{\partial \mathbf{G}} = \boldsymbol{\nu}^* \mathbf{x}^{*T} \qquad \frac{\partial z}{\partial \mathbf{h}} = -\boldsymbol{\nu}^{*T}.$$

When $\ell$ is AOE loss, by chain rule we can multiply each quantity by $\frac{\partial \ell}{\partial z} = \text{sign}(z)$ to get the needed Jacobians. Gradients $\frac{\partial z}{\partial \mathbf{b}}$ and $\frac{\partial z}{\partial \mathbf{h}}$ for the right-hand sides are already well-known as *shadow prices*. If $\mathbf{x}^*$ is degenerate then the relationship between shadow prices and dual variables breaks down, resulting in two-sided shadow prices [Strum, 1969, Aucamp and Steinberg, 1982].

We use degeneracy in the sense of Tijssen and Sierksma [1998] (see Appendix), where a point on the relative interior of the optimal face need not be degenerate, even if there exists a degenerate vertex on the optimal face. This matters when $\mathbf{x}^*$ is non-unique because interior point methods typically converge to the analytical center of the relative interior of the optimal face [Zhang, 1994]. Tijssen and Sierskma also give relations between degeneracy of $\mathbf{x}^*$ and uniqueness of $\boldsymbol{\lambda}^*, \boldsymbol{\nu}^*$, which we apply in Corollary 1. When the gradients are non-unique, this corresponds to the subdifferentiable case.

**Corollary 1.** *In Theorem 1, both* $\frac{\partial z}{\partial \mathbf{b}}$ *and* $\frac{\partial z}{\partial \mathbf{h}}$ *are unique,* $\frac{\partial z}{\partial \mathbf{c}}$ *is unique if and only if* $\mathbf{x}^*$ *is unique, and both* $\frac{\partial z}{\partial \mathbf{A}}$ *and* $\frac{\partial z}{\partial \mathbf{G}}$ *are unique if and only if* $\mathbf{x}^*$ *is unique or* $\mathbf{c} = \mathbf{0}$.

## 4   Experiments

We evaluate our approach by learning a range of synthetic LPs and parametric instances of minimum-cost multi-commodity flow problems. Use of synthetic instances is common in IO (e.g., Ahuja and Orlin [2001], Keshavarz et al. [2011], Dong et al. [2018]) and there are no community-established and readily-available benchmarks, especially for more general formulations. Our experimental study considers instances not directly addressable by previous IO work, either because we learn all coefficients jointly or because the parametrization results in non-convex NLP.

We compare four versions of our gradient-based method ($\text{SQP}_{\text{bprop}}$, $\text{SQP}_{\text{impl}}$, $\text{SQP}_{\text{dir}}$, $\text{SQP}_{\text{cvx}}$) with two gradient-free methods: random search (RS) and COBYLA. A gradient-free baseline is applicable to (ILOP) only if it (i) supports general bilevel natively, or (ii) allows the objective and constraints to be specified by callbacks. COBYLA is conceptually similar to SQP and can be readily applied to (ILOP), but many otherwise-powerful solvers such as BARON [Sahinidis, 2017], CPLEX [IBM, 2020] and Gurobi [Gurobi Optimization, 2020] cannot.

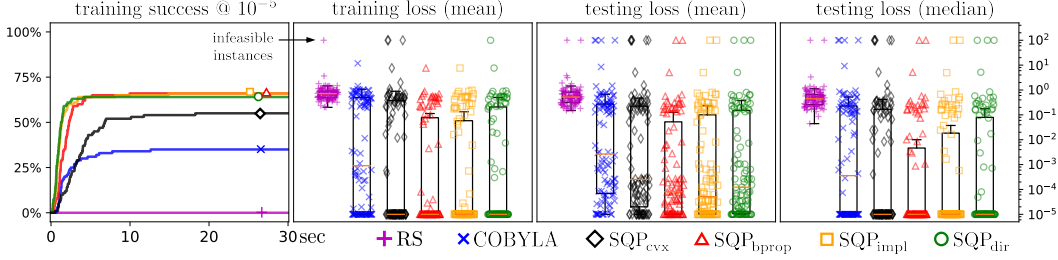

Figure 3: A comparison on synthetic PLP instances having $D=10$ decision variables and $M_1=36$ inequality constraints. Curves show the probability over time of achieving AOE training loss $\ell(\mathbf{w})$ below a tolerance threshold of $10^{-5}$. Box plots show final training and testing loss of 100 different trial instances, each with 20 training and 20 testing points (distinct $\mathbf{u}$ values). We evaluate the "testing loss" for AOE with respect to the 'true' cost $\mathbf{c}(\mathbf{u})$, never the imputed cost. The median loss over a set of testing points tends to be smaller than their mean; see Section 5 for discussion.

Complete experimental results for synthetic LPs are presented in Figures 3, 7, 8, 9. The main observation is that the gradient-based methods perform similarly and become superior to gradient-free methods as the dimension $K$ of parametrization $\mathbf{w}$ increases. We find that including a black-box baseline like COBYLA is important for assessing the practical difficulty of an IO instance (and encourage future papers to do so) because such methods work reasonably well in low-dimensional problems. A second observation is that there are instances for which no method succeeds at minimizing training error 100% of the time. Our method can therefore be viewed as a way to boost the probability of successful training when combined with simple global optimization strategies such as multi-start.

Experiments used PyTorch v1.6 nightly build, the COBYLA and SLSQP wrappers from SciPy v1.4.1, and were run on an Intel Core i7 with 16GB RAM. (We do not use GPUs, though our PyTorch interior point solver inherits GPU acceleration.) We do not regularize $\mathbf{w}$ nor have any other hyperparameters. Code to reproduce experiments is available at `https://github.com/yingcongtan/ilop`.

**Learning linear programs** We used the LP generator of Tan et al. [2019], modifying it to create a more challenging variety of feasible regions; their code did not perform competitively in terms of runtime or success rate on these harder instances, and is not effective at learning constraints under an AOE loss. Fig. 3 shows the task of learning $(\mathbf{c}, \mathbf{A}, \mathbf{b})$ with a $K=6$ dimensional parametrization $\mathbf{w}$, a $D=10$ dimensional decision space $\mathbf{x}$, and 20 training observations. RS fails; COBYLA 'succeeds' on ~30% of instances; $\text{SQP}_{\text{bprop}}, \text{SQP}_{\text{impl}}, \text{SQP}_{\text{dir}}$ succeeds on ~60–65%, which is substantially better. The success curve of $\text{SQP}_{\text{bprop}}$ slightly lags those of $\text{SQP}_{\text{impl}}$ and $\text{SQP}_{\text{dir}}$ due to the overhead of backpropagating through the steps of the interior point solver. $\text{SQP}_{\text{cvx}}$ performs worse in these instances due to the speed at which the internal *scs* solver (written in C) could reach the inner tolerance ($10^{-8}$), and not due to any overhead. See Appendix for five additional problem sizes, where overall the conclusions are the same. On instances with equality constraints, where we learn $(\mathbf{c}, \mathbf{A}, \mathbf{b}, \mathbf{G}, \mathbf{h})$ jointly, performance was similar to the above (see Figure 8 in Appendix).

Much of the IO literature is focused on learning coefficients of $\mathbf{c}$ and/or $\mathbf{b}$ directly, often from a single training target $\mathbf{x}^{\text{obs}}$, i.e., learning a single LP rather than a PLP. In our formulation, we can learn coefficients of $(\mathbf{c}, \mathbf{A}, \mathbf{b}, \mathbf{G}, \mathbf{h})$ jointly by concatenating them into $\mathbf{w}$. For example, an instance with $D=10, M_1=80, M_2=0$ has 890 adjustable parameters. In Figure 9 of Appendix, we show $\text{SQP}_{\text{bprop}}, \text{SQP}_{\text{impl}}$ and $\text{SQP}_{\text{dir}}$ consistently achieve zero AOE training loss on such problems, whereas RS and COBYLA fail to make learning progress given the same time budget. $\text{SQP}_{\text{cvx}}$ makes progress, but is slower than the other gradient-based implementations.

**Learning minimum-cost multi-commodity flow problems** Fig. 4 shows a visualization of our experiment on the Nguyen-Dupuis graph [Nguyen and Dupuis, 1984]. We learn a periodic arc cost $c_j(t, l_j, p_j) = l_j + w_1 p_j + w_2 l_j (\sin(2\pi(w_3 + w_4 t + w_5 l_j)) + 1)$ and an affine arc capacity $b_j(l_j) = 1 + w_6 + w_7 l_j$, based on global feature $t$ (time of day) and arc-specific features $l_j$ (length) and $p_j$ (toll price). To avoid trivial solutions, we set $\mathcal{W} = \{\mathbf{w} \geq \mathbf{0}, w_3 + w_4 + w_5 = 1\}$. Results on 100 instances are shown in Fig. 5. The SQP methods outperform RS and COBYLA in training and testing loss. From an IO perspective the fact that we are jointly learning costs and capacities in a non-convex NLP formulation is already quite general. $\text{SQP}_{\text{cvx}}$ is still slower, but more competitive.

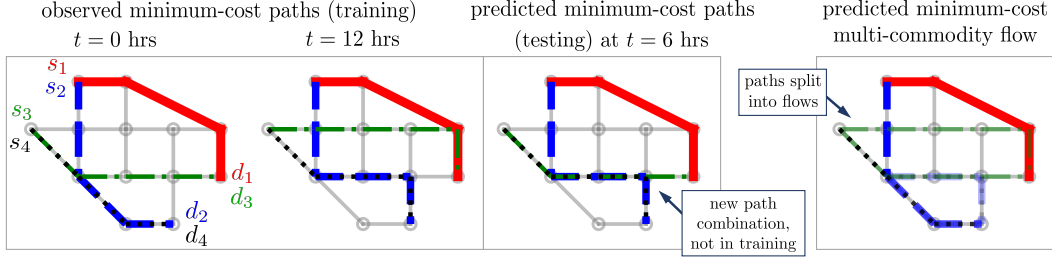

Figure 4: A visualization of minimum-cost paths (for simplicity) and minimum-cost multi-commodity flows (our experiment) on the Nguyen-Dupuis network. Sources $\{s_1, s_2, s_3, s_4\}$ and destinations $\{d_1, d_2, d_3, d_4\}$ are shown. At left are two example sets of training paths $\{(t_1, \mathbf{x}_1^{\text{obs}}), (t_2, \mathbf{x}_2^{\text{obs}})\}$ alongside an example of a correctly predicted set of optimal paths under different conditions (different $t$). At right is a visualization of a correctly predicted optimal flow, where color intensity indicates proportion of flow along arcs.

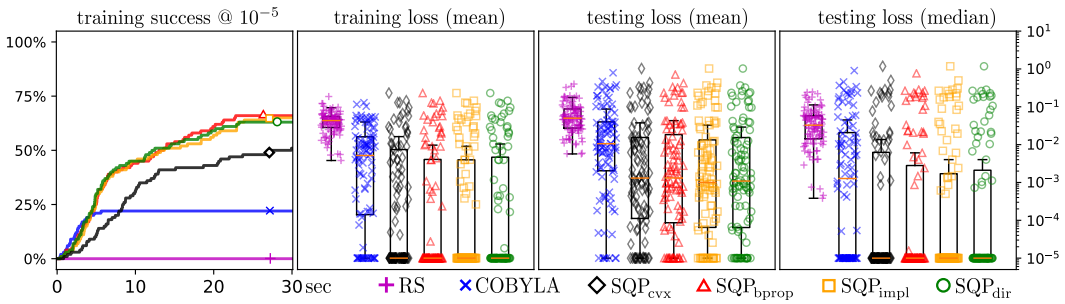

Figure 5: A comparison on minimum-cost multi-commodity flow instances, similar to Fig. 3.

# 5 Discussion

**Generalizing is hard** We report both the mean and median loss over the testing points in each trial. The difference in mean and median testing error is due to the presence of a few 'outliers' among the otherwise-small test set errors. Fig. 6 shows the nature of this failure to generalize: the decision map $\mathbf{u} \mapsto \mathbf{x}^*$ of a PLP has discontinuities, so the training data can easily under-specify the set of learned models that can achieve zero training loss; this is similar to the scenario that motivates the max-margin learning principle, used for good generalization in SVMs. It is not clear what forms of regularization $r(\mathbf{w})$ might reliably improve generalization in IO. Fig. 6 also suggests that training points which closely straddle discontinuities are much more 'valuable' from a learning perspective.

**Scalability** We wish to highlight the scalability of direct gradients, and of our bilevel approach more generally. First, our experiments use a general-purpose LP solver where forward solve dominates runtime. In scenarios where forward solve is fast, for example by an application-specific algorithm

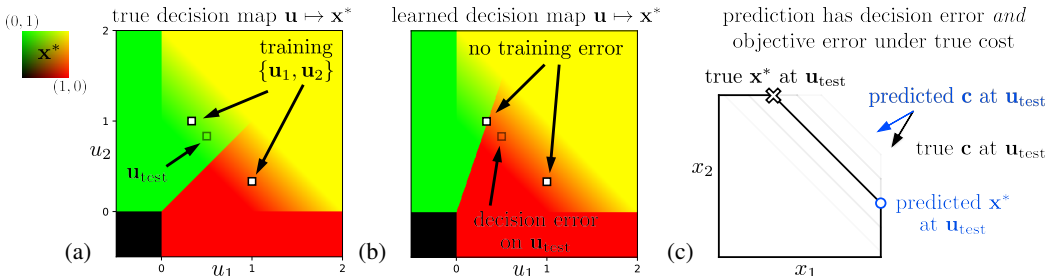

Figure 6: A failure to generalize in a learned PLP. Shown are the optimal decision map $\mathbf{u} \mapsto \mathbf{x}^*$ for a ground-truth PLP (a) and learned PLP (b) with the value of components $(x_1^*, x_2^*)$ represented by red and green intensity respectively, along with that of a PLP trained on $\{\mathbf{u}_1, \mathbf{u}_2\}$. The learned PLP has no training error (SDE$=0$, AOE$=0$) but large test error (SDE$=.89$, AOE$=.22$) as depicted in (c). (See Appendix for the specific PLP used in this example.)

(max-flow, matching, etc.) or a fast re-solve strategy, the gradient computation can be proportionally significant. In that case, direct evaluation of the gradient scales much better than solving backprop or solving the system required by implicit differentiation [Amos and Kolter, 2017, Agrawal et al., 2019]. Shown at right are compute times for an LP parameter gradient averaged over the 100 instances from Fig. 5, with 'direct' being at least ~50x faster than alternatives.

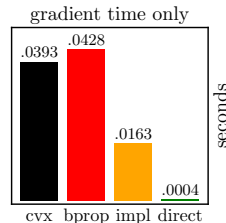

Second, note that other IO approaches often convert a bilevel problem into a new single-level one [Aswani et al., 2018]. This strategy cannot exploit fast algorithms for specialized forward problems (i.e., the inner problem of our formulation) and must rely on general-purpose machinery like CPLEX. By retaining the bilevel nature of (ILOP), our approach allows specialized algorithms to be used for the forward problem, and fast gradients whenever optimal primal and dual solutions can be recovered.

**Generality and applicability**  Our work generalizes in some respects, and specializes others. Considering only literature on inverse *linear* optimization, our (ILOP) formulation generalizes prior work in that it tackles fully parametric PLPs and arbitrary number of observations. Our methodology is meanwhile a novel extension of Tan et al. [2019] since here we introduce 'outer' constraints, SQP-based training, a new gradient computation method, and a faster forward solver implementation. Outside the linear case, our NLP approach can be applied to inverse *convex* optimization because the more general gradient-computation machinery now also exists [Agrawal et al., 2019].

Although we highlighted SQP as a suitable gradient-based NLP solver, other NLP methods may work better in a given setting. Our methodology is applicable for any gradient-based NLP solver allowing specification of objective and constraints via callbacks, thereby being 'agnostic' to the bilevel nature.

## 6    Conclusion

In this paper, we propose a novel bilevel formulation and gradient-based framework for learning linear programs from optimal decisions. The methodology learns all parameters jointly while allowing flexible parametrizations of costs, constraints, and loss functions—a generalization of the problems typically addressed in the inverse linear optimization literature. It furthermore has speed advantages over a gradient-free approach.

Our work allows a strong class of inductive priors, namely parametric linear programs, to be imposed on a hypothesis space for learning. A major motivation for ours and for similar work is that, when the inductive prior is suited to the problem, we can learn a much better (and more interpretable) model, from far less data, than by applying general-purpose machine learning methods. In settings spanning economics, commerce, and healthcare, data on decisions is expensive to obtain and to collect, so we hope that our data-efficient approach will help to build better models and to make better decisions.

## Broader Impact

We believe this work may be of interest to scientists working in machine learning, operations research, mechanism design and/or game theory. The methodology applies whenever desired outcomes of an LP can be given by example, rather than by the LP's coefficients. Linear programs are widely used for planning, for modeling natural phenomena, and for decision-making agents.

A positive impact of this work is faster and more flexible training of LPs. For example, this may be useful for training better recommender systems when users are known to optimize their choices. Another positive impact is data efficiency: LPs are a strong class of inductive priors, so we can learn more interpretable models from less data. Data efficiency is important because collecting data on optimal decisions (and under different conditions) can be very expensive, and because interpretability is important in decision-making settings where unwanted biases may be introduced from the data.

While our goal is for the work to have positive impact, and it has potential applications in co-operative games, negative outcomes are also possible if the framework is applied with malicious or adversarial goals, for example to build a model of an opponent's decision-making process in an adversarial game.

## Acknowledgments and Disclosure of Funding

The authors thank the reviewers for their constructive feedback which improved the paper. This research was supported by the Natural Sciences and Engineering Research Council of Canada.

## Footnotes

[1] NLP constraint vector $\mathbf{h}(\mathbf{w})$ is not the same as FOP right-hand side $\mathbf{h}(\mathbf{u}, \mathbf{w})$, despite same symbol.

[2]In a slight abuse of notation, we ignore leading singleton dimension of $\frac{\partial z}{\partial \mathbf{A}} \in \mathbb{R}^{1 \times M_1 \times D}$, $\frac{\partial z}{\partial \mathbf{G}} \in \mathbb{R}^{1 \times M_2 \times D}$.

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
