[Supplementary Material]

# Appendix

## Appendix A: Forward Optimization Problem for Figure 1

*Forward optimization problem for Figure 1.* The FOP formulation used is shown in (2) below.

$$\begin{aligned}
\underset{x_1,x_2}{\text{minimize}} \quad & \cos(w_1 + w_2 u)x_1 + \sin(w_1 + w_2 u)x_2 \\
\text{subject to} \quad & (1 + w_2 u)x_1 \geq w_1 \\
& (1 + w_1)x_2 \geq w_2 u \\
& x_1 + x_2 \leq 1 + w_1 + w_2 u
\end{aligned} \tag{2}$$

For a fixed $u$ and weights $\mathbf{w} = (w_1, w_2)$ it is an LP. The observation $\mathbf{x}_1^{\text{obs}} = (-0.625, 0.925)$ was generated using $u_1 = 1.0$ with true parameters $\mathbf{w} = (-0.5, -0.2)$.

For illustrative clarity, the panels in Figure 1 depicting the specific feasible regions for $\{\mathbf{w}_1, \mathbf{w}_2, \mathbf{w}_3\}$ are slightly adjusted and stylized from the actual PLP (2), but are qualitatively representative.

## Appendix B: Redundancy Among Target-Feasibility Constraints

Redundant constraints in (1a) are not problematic in principle. Still, removing redundant constraints may help overall performance, either in terms of speed or numerical stability of the 'outer' solver. Here we discuss strategies for automatically removing redundant constraints, depending on assumptions. In this section, when we use $\mathbf{x}$ or $\mathbf{x}_i$ it should be understood to represent some target $\mathbf{x}^{\text{obs}}$ or $\mathbf{x}_i^{\text{obs}}$.

**Constraints that are equivalent.** There may exist indices $i$ and $i'$ for which the corresponding constraints $\mathbf{a}(\mathbf{u}_i, \mathbf{w})^T \mathbf{x}_i \leq b(\mathbf{u}_i, \mathbf{w})$ and $\mathbf{a}(\mathbf{u}_{i'}, \mathbf{w})^T \mathbf{x}_{i'} \leq b(\mathbf{u}_{i'}, \mathbf{w})$ are identical or equivalent. For example, when a constraint is independent of $\mathbf{u}$ this often results in identical training targets $\mathbf{x}_i$ and $\mathbf{x}_{i'}$ that produce identical constraints. The situation for equality constraints is similar.

**Constraints independent of $\mathbf{w}$.** If an individual constraint $\mathbf{a}(\mathbf{u}, \mathbf{w})^T \mathbf{x} \leq b(\mathbf{u}, \mathbf{w})$ is independent of $\mathbf{w}$ then either:

1. $\mathbf{a}(\mathbf{u}_i)^T \mathbf{x}_i \leq b(\mathbf{u}_i)$ for all $i$ so the constraint can be omitted; or,
2. $\mathbf{a}(\mathbf{u}_i)^T \mathbf{x}_i > b(\mathbf{u}_i)$ for some $i$ so the (ILOP) formulation is infeasible due to model misspecification, either in structural assumptions, or assumptions about noise.

The same follows for any equality constraint $\mathbf{g}(\mathbf{u}, \mathbf{w})^T \mathbf{x} = h(\mathbf{u}, \mathbf{w})$ that is independent of $\mathbf{w}$. For example, in our minimum-cost multi-commodity flow experiments, the flow conservation constraints (equality) are independent of $\mathbf{w}$ and so are omitted from (1a) in the corresponding (ILOP) formulation.

**Constraints affinely-dependent in $\mathbf{w}$.** Constraints may be affinely-dependent on parameters $\mathbf{w}$. For example, this is a common assumption in robust optimization [Zhen et al., 2018]. Let $\mathbf{A}(\mathbf{u}, \mathbf{w})$ and $\mathbf{b}(\mathbf{u}, \mathbf{w})$ represent the constraints that are affinely dependent on $\mathbf{w} \in \mathbb{R}^K$. We can write

$$\mathbf{A}(\mathbf{u}, \mathbf{w}) = \mathbf{A}^0(\mathbf{u}) + \sum_{k=1}^{K} w_k \mathbf{A}^k(\mathbf{u}) \qquad \text{and} \qquad \mathbf{b}(\mathbf{u}, \mathbf{w}) = \mathbf{b}^0(\mathbf{u}) + \sum_{k=1}^{K} w_k \mathbf{b}^k(\mathbf{u})$$

for some matrix-valued functions $\mathbf{A}^k(\cdot)$ and vector-valued functions $\mathbf{b}^k(\cdot)$. It is easy to show that we can then rewrite the constraints $\mathbf{A}(\mathbf{u}, \mathbf{w})\mathbf{x} \leq \mathbf{b}(\mathbf{u}, \mathbf{w})$ as $\tilde{\mathbf{A}}(\mathbf{u}, \mathbf{x})\mathbf{w} \leq \tilde{\mathbf{b}}(\mathbf{u}, \mathbf{x})$ where

$$\tilde{\mathbf{A}}(\mathbf{u}, \mathbf{x}) = \begin{bmatrix} \mathbf{A}^1(\mathbf{u})\mathbf{x} - \mathbf{b}^1(\mathbf{u}) & \cdots & \mathbf{A}^K(\mathbf{u})\mathbf{x} - \mathbf{b}^K(\mathbf{u}) \end{bmatrix}$$
$$\tilde{\mathbf{b}}(\mathbf{u}, \mathbf{x}) = \mathbf{b}^0(\mathbf{u}) - \mathbf{A}^0(\mathbf{u})\mathbf{x}.$$

Similarly if $\mathbf{G}(\mathbf{u}, \mathbf{w})\mathbf{x} = \mathbf{h}(\mathbf{u}, \mathbf{w})$ are affine in $\mathbf{w}$ we can rewrite them as $\tilde{\mathbf{G}}(\mathbf{u}, \mathbf{x})\mathbf{w} = \tilde{\mathbf{h}}(\mathbf{u}, \mathbf{x})$. If we apply these functions across all training samples $i = 1, \ldots, N$, and stack their coefficients as

$$\tilde{\mathbf{A}} = \left[\tilde{\mathbf{A}}(\mathbf{u}_i, \mathbf{x}_i)\right]_{i=1}^{N}, \quad \tilde{\mathbf{b}} = \left[\tilde{\mathbf{b}}(\mathbf{u}_i, \mathbf{x}_i)\right]_{i=1}^{N}, \quad \tilde{\mathbf{G}} = \left[\tilde{\mathbf{G}}(\mathbf{u}_i, \mathbf{x}_i)\right]_{i=1}^{N}, \quad \tilde{\mathbf{h}} = \left[\tilde{\mathbf{h}}(\mathbf{u}_i, \mathbf{x}_i)\right]_{i=1}^{N}$$

then the corresponding (ILOP) constraints (1a) reduce to a set of linear 'outer' constraints $\tilde{\mathbf{A}}\mathbf{w} \leq \tilde{\mathbf{b}}$ and $\tilde{\mathbf{G}}\mathbf{w} = \tilde{\mathbf{h}}$ where $\tilde{\mathbf{A}} \in \mathbb{R}^{NM_1 \times K}, \tilde{\mathbf{b}} \in \mathbb{R}^{NM_1}, \tilde{\mathbf{G}} \in \mathbb{R}^{NM_2 \times K}, \tilde{\mathbf{h}} \in \mathbb{R}^{NM_2}$. These reformulated constraint matrices are the system within which we eliminate redundancy in the affinely-dependent case, continued below.

**Equality constraints affinely-dependent in w.** We can eliminate affinely-dependent equality constraint sets by reparametrizing the (ILOP) search over a lower-dimensional space; this is what we do for the experiments with equality constraints shown in Figure 8, although the conclusions do not change with or without this reparametrization. To reparametrize the (ILOP) problem, compute a Moore-Penrose pseudoinverse $\tilde{\mathbf{G}}^+ \in \mathbb{R}^{K \times NM_2}$ to get a direct parametrization of constrained vector $\mathbf{w}$ in terms of an unconstrained vector $\mathbf{w}' \in \mathbb{R}^K$:

$$\mathbf{w}(\mathbf{w}') = \tilde{\mathbf{G}}^+\tilde{\mathbf{h}} + (\mathbf{I} - \tilde{\mathbf{G}}^+\tilde{\mathbf{G}})\mathbf{w}'. \tag{3}$$

By reparametrizing (ILOP) in terms of $\mathbf{w}'$ we guarantee $\tilde{\mathbf{G}}\mathbf{w}(\mathbf{w}') = \tilde{\mathbf{h}}$ is satisfied and can drop equality constraints from (1a) entirely. There are three practical issues with (3):

1. Constrained vector $\mathbf{w}$ only has $K' \equiv K - \text{rank}(\tilde{\mathbf{G}})$ degrees of freedom, so we would like to re-parametrize over a lower-dimensional $\mathbf{w}' \in \mathbb{R}^{K'}$.

2. To search over $\mathbf{w}' \in \mathbb{R}^{K'}$ we need to specify $\tilde{\mathbf{A}}' \in \mathbb{R}^{NM_1 \times K'}$ and $\tilde{\mathbf{b}}' \in \mathbb{R}^{NM_1}$ such that $\tilde{\mathbf{A}}'\mathbf{w}' \leq \tilde{\mathbf{b}}'$ is equivalent to $\tilde{\mathbf{A}}\mathbf{w}(\mathbf{w}') \leq \tilde{\mathbf{b}}$.

3. Given initial $\mathbf{w}_{\text{ini}} \in \mathbb{R}^K$ we need a corresponding $\mathbf{w}'_{\text{ini}} \in \mathbb{R}^{K'}$ to initialize our search.

To address the first issue, we can let the final $K - K'$ components of $\mathbf{w}' \in \mathbb{R}^K$ in (3) be zero, which corresponds to using a lower-dimensional $\mathbf{w}' \in \mathbb{R}^{K'}$. As shorthand let matrix $\mathbf{P} \in \mathbb{R}^{K \times K'}$ be

$$\mathbf{P} \equiv (\mathbf{I}_{K \times K} - \tilde{\mathbf{G}}^+\tilde{\mathbf{G}})\mathbf{I}_{K \times K'} = \mathbf{I}_{K \times K'} - (\tilde{\mathbf{G}}^+\tilde{\mathbf{G}})_{1:K,1:K'}$$

where $\mathbf{I}_{K \times K'}$ denotes $\begin{bmatrix} \mathbf{I}_{K' \times K'} \\ \mathbf{0}_{(K-K') \times K'} \end{bmatrix}$ as in `torch.eye(K, K')` and $(\mathbf{G}^+\mathbf{G})_{1:K,1:K'}$ denotes the first $K'$ columns of $K \times K$ matrix $\mathbf{G}^+\mathbf{G}$. Then we have $\mathbf{w}(\mathbf{w}') = \mathbf{G}^+\mathbf{h} + \mathbf{P}\mathbf{w}'$ where the full dimension of $\mathbf{w}' \in \mathbb{R}^{K'}$ matches the degrees of freedom in $\mathbf{w}$ subject to $\tilde{\mathbf{G}}\mathbf{w} = \tilde{\mathbf{h}}$ and we have $\tilde{\mathbf{G}}\mathbf{w}(\mathbf{w}') = \tilde{\mathbf{h}}$ for any choice of $\mathbf{w}'$.

To address the second issue, simplifying $\tilde{\mathbf{A}}\mathbf{w}(\mathbf{w}') \leq \tilde{\mathbf{b}}$ gives inequality constraints $\tilde{\mathbf{A}}'\mathbf{w}' \leq \tilde{\mathbf{b}}'$ with $\tilde{\mathbf{A}}' = \tilde{\mathbf{A}}\mathbf{P}$ and $\tilde{\mathbf{b}}' = \tilde{\mathbf{b}} - \tilde{\mathbf{A}}\tilde{\mathbf{G}}^+\tilde{\mathbf{h}}$.

To address the third issue we must solve for $\mathbf{w}'_{\text{ini}} \in \mathbb{R}^{K'}$ in the linear system $\mathbf{P}\mathbf{w}'_{\text{ini}} = \mathbf{w}_{\text{ini}} - \tilde{\mathbf{G}}^+\tilde{\mathbf{h}}$. Since $\text{rank}(\mathbf{P}) = K'$ the solution exists and is unique.

Consider also the effect of this reparametrization when $\tilde{\mathbf{G}}\mathbf{w} = \tilde{\mathbf{h}}$ is an infeasible system, for example due to noisy observations or misspecified constraints. In that case searching over $\mathbf{w}'$ automatically restricts the search to $\mathbf{w}$ that satisfy $\tilde{\mathbf{G}}\mathbf{w} = \tilde{\mathbf{h}}$ in a least squares sense, akin to adding an infinitely-weighted $\|\tilde{\mathbf{G}}\mathbf{w} - \tilde{\mathbf{h}}\|^2$ term to the (ILOP) objective.

**Inequality constraints affinely-dependent in w.** After transforming affinely-dependent inequality constraints to $\tilde{\mathbf{A}}'\mathbf{w}' \leq \tilde{\mathbf{b}}'$, detecting redundancy among these constraints can be as hard as solving an LP [Telgen, 1983]. Generally, inequality constraint $\mathbf{a}_j^T\mathbf{w} \leq b_j$ is redundant with respect to $\mathbf{A}\mathbf{w} \leq \mathbf{b}$ if and only if the optimal value of the following LP is non-negative:

$$\begin{aligned} \underset{\mathbf{w}}{\text{minimize}} \quad & b_j - \mathbf{a}_j^T\mathbf{w} \\ \text{subject to} \quad & \mathbf{A}_{\{j' \neq j\}}\mathbf{w} \leq \mathbf{b}_{\{j' \neq j\}} \end{aligned} \tag{4}$$

Here $\mathbf{a}_j$ is the $j^{\text{th}}$ row of $\mathbf{A}$ and $\mathbf{A}_{\{j' \neq j\}}$ is all the rows of $\mathbf{A}$ except the $j^{\text{th}}$. If the optimal value to (4) is non-negative then it says "we tried to violate the $j^{\text{th}}$ constraint, but the other constraints prevented it, and so the $j^{\text{th}}$ constraint must be redundant." However, Telgen [1983] reviews much more efficient methods of identifying redundant linear inequality constraints, by analysis of basic basic variables in a simplex tableau. Zhen et al. [2018] proposed a 'redundant constraint identification' (RCI) procedure that is directly analogous to (4) along with another heuristic RCI procedure.

**Constraints polynomially-dependent in w.** Similar to the affinely-dependent case, when the coefficients of constraints $\mathbf{A}(\mathbf{u}, \mathbf{w})\mathbf{x} \leq \mathbf{b}(\mathbf{u}, \mathbf{w})$ and $\mathbf{G}(\mathbf{u}, \mathbf{w})\mathbf{x} \leq \mathbf{h}(\mathbf{u}, \mathbf{w})$ are polynomially-dependent on $\mathbf{w}$, we can rewrite the constraints in terms of $\mathbf{w}$. Redundancy among equality constraints of the resulting system can be simplified by computing a minimal Gröbner basis [Cox et al., 2013], for example by Buchberger's algorithm which is a generalization of Gaussian elimination; see the paper by Lim and Brunner [2012] for a review of Gröbner basis techniques applicable over a real

field. Redundancy among inequality constraints for nonlinear programming has been studied [Caron, 2009, Obuchowska and Caron, 1995]. Simplifying polynomial systems of equalities and inequalities is a subject of semialgebraic geometry and involves generalizations of Fourier-Motzkin elimination. Details are beyond the scope of this manuscript.

**Appendix C: Proofs of Theorem 1 and Corollary 1**

Degeneracy is often defined for vertices, but solutions returned by an interior point solver tend toward the analytical center of the optimal face. We first define non-degeneracy for a face of an LP model, following Tijssen and Sierksma [1998] and Sierksma and Tijssen [2003]. We then use this definition to define non-degeneracy of a particular solution on the optimal face of an LP model.

Let $F$ be a face of the polyhedron $P$. A constraint of $P$ is *binding* on $F$ if it is satisfied with equality for every point of $F$. Let $\dim(F)$ and $n$ denote the dimension of $F$ and $P$ respectively and $\mathrm{bnd}(F, P)$ denote the number of constraints of $P$ that are binding on $F$.

**Definition 1** (Tijssen & Sierksma 1998). *The* degeneracy degree *of a face $F$ with respect to polyhedron $P$ is $\sigma(F, P) = \mathrm{bnd}(F, P) + \dim(F) - n$.*

**Definition 2** (Tijssen & Sierksma 1998). *A face $F$ of polyhedron $P$ is* degenerate *iff $\sigma(F, P) > 0$, and* non-degenerate *iff $\sigma(F, P) = 0$.*

**Definition 3.** *Given an LP with feasible set $P$, an optimal solution $\mathbf{x}^*$ is* non-degenerate *iff the smallest face $F$ containing $\mathbf{x}^*$ is non-degenerate, i.e., $\sigma(F, P) = 0$ for the smallest face with $\mathbf{x}^* \in F$.*

By these definitions, a solution on the relative interior of the optimal face may be non-degenerate, even when other sub-faces of the optimal face (including vertices) are degenerate in the usual sense.

To assist the proof of Theorem 1, we first show that the following lemma is true.

**Lemma 1.** *If $\mathbf{x}^*$ is a non-degenerate solution to an LP, then the constraints active at $\mathbf{x}^*$ are linearly independent.*

*Proof.* Let $P$ denote the feasible set of the LP, and let $F$ denote the smallest face containing $\mathbf{x}^* \in \mathbb{R}^n$. Let $\{\mathbf{a}_1, \ldots, \mathbf{a}_k\}$ denote the set of constraints binding on $F$, so that $\mathrm{bnd}(F, P) = k$. By non-degeneracy of $\mathbf{x}^*$ we have $\sigma(F, P) = 0$ and so $k = n - \dim(F)$. Since $\dim(F) + \mathrm{rank}\{\mathbf{a}_1, \ldots, \mathbf{a}_k\} = n$ must also hold, we have $\mathrm{rank}\{\mathbf{a}_1, \ldots, \mathbf{a}_k\} = k$, i.e., the constraints binding face $F$ are of full rank.

Now consider whether a constraint $\mathbf{a}_{k+1}$ can be active at $\mathbf{x}^*$ but not binding on $F$. Since $\mathbf{a}_{k+1}$ is not binding on $F$, it must be must be linearly independent from the constraints that are binding on $F$, i.e., $\mathrm{rank}\{\mathbf{a}_1, \ldots, \mathbf{a}_k, \mathbf{a}_{k+1}\} = k + 1$. But if $\mathbf{a}_{k+1}$ were also active at $\mathbf{x}^*$, this would imply the existence of a face of $P$ containing $\mathbf{x}^*$ and having dimension $n - k - 1$. Since $F$ has dimension $n - k$, this would contradict our assumption that $F$ is the smallest face containing $\mathbf{x}^*$. Therefore $\{\mathbf{a}_1, \ldots, \mathbf{a}_k\}$ must comprise all constraints that are active on $\mathbf{x}^*$, and they are linearly independent. $\square$

*Proof of Theorem 1.* The dual linear program associated with (LP) is

$$\begin{aligned}
\underset{\boldsymbol{\lambda}, \boldsymbol{\nu}}{\text{maximize}} \quad & \mathbf{b}^T \boldsymbol{\lambda} + \mathbf{h}^T \boldsymbol{\nu} \\
\text{subject to} \quad & \mathbf{A}^T \boldsymbol{\lambda} + \mathbf{G}^T \boldsymbol{\nu} = \mathbf{c} \\
& \boldsymbol{\lambda} \leq \mathbf{0},
\end{aligned} \quad \text{(DP)}$$

where $\boldsymbol{\lambda} \in \mathbb{R}^{M_1}_{\leq 0}, \boldsymbol{\nu} \in \mathbb{R}^{M_2}$ are the associated dual variables for the primal inequality and equality constraints, respectively.

Since $\mathbf{x}^*$ is optimal to (LP) and $\boldsymbol{\lambda}^*, \boldsymbol{\nu}^*$ are optimal to (DP), then $(\mathbf{x}^*, \boldsymbol{\lambda}^*, \boldsymbol{\nu}^*)$ satisfy the KKT conditions (written specialized to the particular LP form we use):

$$\begin{aligned}
\mathbf{A}\mathbf{x} &\leq \mathbf{b} \\
\mathbf{G}\mathbf{x} &= \mathbf{h} \\
\mathbf{A}^T \boldsymbol{\lambda} + \mathbf{G}^T \boldsymbol{\nu} &= \mathbf{c} \\
\boldsymbol{\lambda} &\leq \mathbf{0} \\
\mathbf{D}(\boldsymbol{\lambda})(\mathbf{A}\mathbf{x} - \mathbf{b}) &= \mathbf{0}
\end{aligned} \quad \text{(KKT)}$$

where $\mathbf{D}(\boldsymbol{\lambda})$ is the diagonal matrix having $\boldsymbol{\lambda}$ on the diagonal. The first two constraints correspond to primal feasibility, the next two to dual feasibility and the last one specifies complementary slackness. From here forward it should be understood that $\mathbf{x}, \boldsymbol{\lambda}, \boldsymbol{\nu}$ satisfy KKT even when not emphasized by $*$.

As in the paper by Amos and Kolter [2017], implicitly differentiating the equality constraints in (KKT) gives

$$
\begin{aligned}
\mathbf{G}\mathrm{d}\mathbf{x} &= \mathrm{d}\mathbf{h} - \mathrm{d}\mathbf{G}\mathbf{x} \\
\mathbf{A}^T\mathrm{d}\boldsymbol{\lambda} + \mathbf{G}^T\mathrm{d}\boldsymbol{\nu} &= \mathrm{d}\mathbf{c} - \mathrm{d}\mathbf{A}^T\boldsymbol{\lambda} - \mathrm{d}\mathbf{G}^T\boldsymbol{\nu} \\
\mathbf{D}(\boldsymbol{\lambda})\mathbf{A}\mathrm{d}\mathbf{x} + \mathbf{D}(\mathbf{A}\mathbf{x} - \mathbf{b})\mathrm{d}\boldsymbol{\lambda} &= \mathbf{D}(\boldsymbol{\lambda})(\mathrm{d}\mathbf{b} - \mathrm{d}\mathbf{A}\mathbf{x})
\end{aligned}
\tag{DKKT}
$$

where $\mathrm{d}\mathbf{c}, \mathrm{d}\mathbf{A}, \mathrm{d}\mathbf{b}, \mathrm{d}\mathbf{G}, \mathrm{d}\mathbf{h}$ are parameter differentials and $\mathrm{d}\mathbf{x}, \mathrm{d}\boldsymbol{\lambda}, \mathrm{d}\boldsymbol{\nu}$ are solution differentials, all having the same dimensions as the variables they correspond to. Because (KKT) is a second-order system, (DKKT) is a system of linear equations. Because the system is linear, a partial derivative such as $\frac{\partial x_j^*}{\partial b_i}$ can be determined (if it exists) by setting $\mathrm{d}b_i = 1$ and all other parameter differentials to $0$, then solving the system for solution differential $\mathrm{d}x_j$, as shown by Amos and Kolter [2017].

We can assume (KKT) is feasible in $\mathbf{x}, \boldsymbol{\lambda}, \boldsymbol{\nu}$. In each case of the main proof it will be important to characterize conditions under which (DKKT) is then feasible in $\mathrm{d}\mathbf{x}$. This is because, if (DKKT) is feasible in at least $\mathrm{d}\mathbf{x}$, then by substitution we have

$$
\begin{aligned}
\mathbf{c}^T\mathrm{d}\mathbf{x} &= (\mathbf{A}^T\boldsymbol{\lambda} + \mathbf{G}^T\boldsymbol{\nu})^T\mathrm{d}\mathbf{x} \\
&= \boldsymbol{\lambda}^T\mathbf{A}\mathrm{d}\mathbf{x} + \boldsymbol{\nu}^T\mathbf{G}\mathrm{d}\mathbf{x} \\
&= \boldsymbol{\lambda}^T(\mathrm{d}\mathbf{b} - \mathrm{d}\mathbf{A}\mathbf{x}) + \boldsymbol{\nu}^T(\mathrm{d}\mathbf{h} - \mathrm{d}\mathbf{G}\mathbf{x})
\end{aligned}
\tag{5}
$$

and this substitution is what gives the total derivatives their form. In (5) the substitution $\boldsymbol{\lambda}^T\mathbf{A}\mathrm{d}\mathbf{x} = \boldsymbol{\lambda}^T(\mathrm{d}\mathbf{b} - \mathrm{d}\mathbf{A}\mathbf{x})$ holds because $\mathbf{x}, \boldsymbol{\lambda}$ feasible in (KKT) implies $\lambda_i < 0 \Rightarrow \mathbf{A}_i\mathbf{x} - b_i = 0$ in (DKKT), where $\mathbf{A}_i$ is the $i^{\text{th}}$ row of $\mathbf{A}$. Whenever $\mathrm{d}\mathbf{x}$ is feasible in (DKKT) we have $\lambda_i\mathbf{A}_i\mathrm{d}\mathbf{x} = \lambda_i(\mathrm{d}b_i - \mathrm{d}\mathbf{A}_i\mathbf{x})$ for any $\lambda_i \leq 0$, where $\mathrm{d}\mathbf{A}_i$ is the $i^{\text{th}}$ row of differential $\mathrm{d}\mathbf{A}$.

Note that (5) holds even if (DKKT) is not feasible in $\mathrm{d}\boldsymbol{\lambda}$ and/or $\mathrm{d}\boldsymbol{\nu}$. In other words, it does not require the KKT point $(\mathbf{x}^*, \boldsymbol{\lambda}^*, \boldsymbol{\nu}^*)$ to be differentiable with respect to $\boldsymbol{\lambda}^*$ and/or $\boldsymbol{\nu}^*$.

Given a KKT point $(\mathbf{x}^*, \boldsymbol{\lambda}^*, \boldsymbol{\nu}^*)$ let $\mathcal{I}, \mathcal{J}, \mathcal{K}$ be a partition of inequality indices $\{1, \ldots, M_1\}$ where

$$
\begin{aligned}
\mathcal{I} &= \{\, i : \lambda_i^* < 0, \ \mathbf{A}_i\mathbf{x}^* = b_i \,\} \\
\mathcal{J} &= \{\, i : \lambda_i^* = 0, \ \mathbf{A}_i\mathbf{x}^* < b_i \,\} \\
\mathcal{K} &= \{\, i : \lambda_i^* = 0, \ \mathbf{A}_i\mathbf{x}^* = b_i \,\}
\end{aligned}
$$

and the corresponding submatrices of $\mathbf{A}$ are $\mathbf{A}_{\mathcal{I}}, \mathbf{A}_{\mathcal{J}}, \mathbf{A}_{\mathcal{K}}$. Then (DKKT) in matrix form is

$$
\begin{bmatrix}
\mathbf{G} & \mathbf{0} & \mathbf{0} & \mathbf{0} & \mathbf{0} \\
\mathbf{D}(\boldsymbol{\lambda}_{\mathcal{I}})\mathbf{A}_{\mathcal{I}} & \mathbf{0} & \mathbf{0} & \mathbf{0} & \mathbf{0} \\
\mathbf{0} & \mathbf{0} & \mathbf{D}(\mathbf{A}_{\mathcal{J}}\mathbf{x} - \mathbf{b}_{\mathcal{J}}) & \mathbf{0} & \mathbf{0} \\
\mathbf{0} & \mathbf{0} & \mathbf{0} & \mathbf{0} & \mathbf{0} \\
\mathbf{0} & \mathbf{A}_{\mathcal{I}}^T & \mathbf{A}_{\mathcal{J}}^T & \mathbf{A}_{\mathcal{K}}^T & \mathbf{G}^T
\end{bmatrix}
\begin{bmatrix}
\mathrm{d}\mathbf{x} \\
\mathrm{d}\boldsymbol{\lambda}_{\mathcal{I}} \\
\mathrm{d}\boldsymbol{\lambda}_{\mathcal{J}} \\
\mathrm{d}\boldsymbol{\lambda}_{\mathcal{K}} \\
\mathrm{d}\boldsymbol{\nu}
\end{bmatrix}
=
\begin{bmatrix}
\mathrm{d}\mathbf{h} - \mathrm{d}\mathbf{G}\mathbf{x} \\
\mathrm{d}\mathbf{b}_{\mathcal{I}} - \mathrm{d}\mathbf{A}_{\mathcal{I}}\mathbf{x} \\
\mathbf{0} \\
\mathbf{0} \\
\mathrm{d}\mathbf{c} - \mathrm{d}\mathbf{A}^T\boldsymbol{\lambda} - \mathrm{d}\mathbf{G}^T\boldsymbol{\nu}
\end{bmatrix}
\tag{6}
$$

The pattern of the proof in each case will be to characterize feasibility of (6) in $\mathrm{d}\mathbf{x}$ and then apply (5) for the result.

**Evaluating $\frac{\partial z}{\partial \mathbf{c}}$.** Consider $\frac{\partial z}{\partial c_j} = x_j^{\text{obs}} - x_j^* - \mathbf{c}^T\frac{\partial \mathbf{x}^*}{\partial c_j}$. To evaluate the $\mathbf{c}^T\frac{\partial \mathbf{x}^*}{\partial c_j}$ term, set $\mathrm{d}c_j = 1$ and all other parameter differentials to $0$. Then the right-hand side of (6) becomes

$$
\begin{bmatrix}
\mathbf{G} & \mathbf{0} & \mathbf{0} & \mathbf{0} & \mathbf{0} \\
\mathbf{D}(\boldsymbol{\lambda}_{\mathcal{I}})\mathbf{A}_{\mathcal{I}} & \mathbf{0} & \mathbf{0} & \mathbf{0} & \mathbf{0} \\
\mathbf{0} & \mathbf{0} & \mathbf{D}(\mathbf{A}_{\mathcal{J}}\mathbf{x} - \mathbf{b}_{\mathcal{J}}) & \mathbf{0} & \mathbf{0} \\
\mathbf{0} & \mathbf{0} & \mathbf{0} & \mathbf{0} & \mathbf{0} \\
\mathbf{0} & \mathbf{A}_{\mathcal{I}}^T & \mathbf{A}_{\mathcal{J}}^T & \mathbf{A}_{\mathcal{K}}^T & \mathbf{G}^T
\end{bmatrix}
\begin{bmatrix}
\mathrm{d}\mathbf{x} \\
\mathrm{d}\boldsymbol{\lambda}_{\mathcal{I}} \\
\mathrm{d}\boldsymbol{\lambda}_{\mathcal{J}} \\
\mathrm{d}\boldsymbol{\lambda}_{\mathcal{K}} \\
\mathrm{d}\boldsymbol{\nu}
\end{bmatrix}
=
\begin{bmatrix}
\mathbf{0} \\
\mathbf{0} \\
\mathbf{0} \\
\mathbf{0} \\
\mathbf{1}^j
\end{bmatrix}
\tag{7}
$$

where $\mathbf{1}^j$ denotes the vector with 1 for component $j$ and 0 elsewhere. System (7) is feasible in $\mathrm{d}\mathbf{x}$ (not necessarily unique) so we can apply (5) to get $\mathbf{c}^T \frac{\partial \mathbf{x}^*}{\partial c_j} = \mathbf{c}^T \mathrm{d}\mathbf{x} = \boldsymbol{\lambda}^T(\mathbf{0} - \mathbf{0}\mathbf{x}) + \boldsymbol{\nu}^T(\mathbf{0} - \mathbf{0}\mathbf{x}) = 0$. The result for $\frac{\partial z}{\partial \mathbf{c}}$ then follows from $\mathbf{c}^T \frac{\partial \mathbf{x}^*}{\partial \mathbf{c}} = \mathbf{0}$.

**Evaluating $\frac{\partial z}{\partial \mathbf{h}}$.** Consider $\frac{\partial z}{\partial h_i} = -\mathbf{c}^T \frac{\partial \mathbf{x}^*}{\partial h_i}$. Set $\mathrm{d}h_i = 1$ and all other parameter differentials to 0. Then the right-hand side of (6) becomes

$$
\begin{bmatrix}
\mathbf{G} & \mathbf{0} & \mathbf{0} & \mathbf{0} & \mathbf{0} \\
\mathbf{D}(\boldsymbol{\lambda}_{\mathcal{I}})\mathbf{A}_{\mathcal{I}} & \mathbf{0} & \mathbf{0} & \mathbf{0} & \mathbf{0} \\
\mathbf{0} & \mathbf{0} & \mathbf{D}(\mathbf{A}_{\mathcal{J}}\mathbf{x} - \mathbf{b}_{\mathcal{J}}) & \mathbf{0} & \mathbf{0} \\
\mathbf{0} & \mathbf{0} & \mathbf{0} & \mathbf{0} & \mathbf{0} \\
\mathbf{0} & \mathbf{A}_{\mathcal{I}}^T & \mathbf{A}_{\mathcal{J}}^T & \mathbf{A}_{\mathcal{K}}^T & \mathbf{G}^T
\end{bmatrix}
\begin{bmatrix}
\mathrm{d}\mathbf{x} \\
\mathrm{d}\boldsymbol{\lambda}_{\mathcal{I}} \\
\mathrm{d}\boldsymbol{\lambda}_{\mathcal{J}} \\
\mathrm{d}\boldsymbol{\lambda}_{\mathcal{K}} \\
\mathrm{d}\boldsymbol{\nu}
\end{bmatrix}
=
\begin{bmatrix}
\mathbf{1}^i \\
\mathbf{0} \\
\mathbf{0} \\
\mathbf{0} \\
\mathbf{0}
\end{bmatrix}
\tag{8}
$$

Since $\mathbf{x}^*$ is non-degenerate in the sense of Definition 3, then there are at most $D$ active constraints (including equality constraints) and by Lemma 1 the rows of $\begin{bmatrix} \mathbf{G} \\ \mathbf{A}_{\mathcal{I}} \end{bmatrix}$ are also linearly independent. Since active constraints are linearly independent, system (8) is feasible in $\mathrm{d}\mathbf{x}$ across all $i \in \{1, \ldots, M_2\}$. We can therefore apply (5) to get $\mathbf{c}^T \frac{\partial \mathbf{x}^*}{\partial h_i} = \mathbf{c}^T \mathrm{d}\mathbf{x} = \boldsymbol{\lambda}^T(\mathbf{0} - \mathbf{0}\mathbf{x}) + \boldsymbol{\nu}^T(\mathbf{1}^i - \mathbf{0}\mathbf{x}) = \nu_i$. The result for $\frac{\partial z}{\partial \mathbf{h}}$ then follows from $\mathbf{c}^T \frac{\partial \mathbf{x}^*}{\partial \mathbf{h}} = \boldsymbol{\nu}^{*T}$.

**Evaluating $\frac{\partial z}{\partial \mathbf{b}}$.** Consider $\frac{\partial z}{\partial b_i} = -\mathbf{c}^T \frac{\partial \mathbf{x}^*}{\partial b_i}$. Set $\mathrm{d}b_i = 1$ and all other parameter differentials to 0. For $i \in \mathcal{I}$ the right-hand side of (6) becomes

$$
\begin{bmatrix}
\mathbf{G} & \mathbf{0} & \mathbf{0} & \mathbf{0} & \mathbf{0} \\
\mathbf{D}(\boldsymbol{\lambda}_{\mathcal{I}})\mathbf{A}_{\mathcal{I}} & \mathbf{0} & \mathbf{0} & \mathbf{0} & \mathbf{0} \\
\mathbf{0} & \mathbf{0} & \mathbf{D}(\mathbf{A}_{\mathcal{J}}\mathbf{x} - \mathbf{b}_{\mathcal{J}}) & \mathbf{0} & \mathbf{0} \\
\mathbf{0} & \mathbf{0} & \mathbf{0} & \mathbf{0} & \mathbf{0} \\
\mathbf{0} & \mathbf{A}_{\mathcal{I}}^T & \mathbf{A}_{\mathcal{J}}^T & \mathbf{A}_{\mathcal{K}}^T & \mathbf{G}^T
\end{bmatrix}
\begin{bmatrix}
\mathrm{d}\mathbf{x} \\
\mathrm{d}\boldsymbol{\lambda}_{\mathcal{I}} \\
\mathrm{d}\boldsymbol{\lambda}_{\mathcal{J}} \\
\mathrm{d}\boldsymbol{\lambda}_{\mathcal{K}} \\
\mathrm{d}\boldsymbol{\nu}
\end{bmatrix}
=
\begin{bmatrix}
\mathbf{0} \\
\lambda_i \mathbf{1}^i \\
\mathbf{0} \\
\mathbf{0} \\
\mathbf{0}
\end{bmatrix}
\tag{9}
$$

Since $\mathbf{x}^*$ is non-degenerate, then system (9) is feasible in $\mathrm{d}\mathbf{x}$ for all $i \in \mathcal{I}$ by identical reasoning as for $\frac{\partial z}{\partial h_i}$. For $i \in \mathcal{J} \cup \mathcal{K}$ the right-hand side of (6) is zero and so the system is feasible in $\mathrm{d}\mathbf{x}$. System (9) is therefore feasible in $\mathrm{d}\mathbf{x}$ across all $i \in \{1, \ldots, M_1\}$. We can therefore apply (5) to get $\mathbf{c}^T \frac{\partial \mathbf{x}^*}{\partial b_i} = \mathbf{c}^T \mathrm{d}\mathbf{x} = \boldsymbol{\lambda}^T(\mathbf{1}^i - \mathbf{0}\mathbf{x}) + \boldsymbol{\nu}^T(\mathbf{0} - \mathbf{0}\mathbf{x}) = \lambda_i$. The result for $\frac{\partial z}{\partial \mathbf{b}}$ then follows from $\mathbf{c}^T \frac{\partial \mathbf{x}^*}{\partial \mathbf{b}} = \boldsymbol{\lambda}^{*T}$.

**Evaluating $\frac{\partial z}{\partial \mathbf{G}}$.** Consider $\frac{\partial z}{\partial G_{ij}} = -\mathbf{c}^T \frac{\partial \mathbf{x}^*}{\partial G_{ij}}$. Set $\mathrm{d}G_{ij} = 1$ and all other parameter differentials to 0. Then the right-hand side of (6) becomes

$$
\begin{bmatrix}
\mathbf{G} & \mathbf{0} & \mathbf{0} & \mathbf{0} & \mathbf{0} \\
\mathbf{D}(\boldsymbol{\lambda}_{\mathcal{I}})\mathbf{A}_{\mathcal{I}} & \mathbf{0} & \mathbf{0} & \mathbf{0} & \mathbf{0} \\
\mathbf{0} & \mathbf{0} & \mathbf{D}(\mathbf{A}_{\mathcal{J}}\mathbf{x} - \mathbf{b}_{\mathcal{J}}) & \mathbf{0} & \mathbf{0} \\
\mathbf{0} & \mathbf{0} & \mathbf{0} & \mathbf{0} & \mathbf{0} \\
\mathbf{0} & \mathbf{A}_{\mathcal{I}}^T & \mathbf{A}_{\mathcal{J}}^T & \mathbf{A}_{\mathcal{K}}^T & \mathbf{G}^T
\end{bmatrix}
\begin{bmatrix}
\mathrm{d}\mathbf{x} \\
\mathrm{d}\boldsymbol{\lambda}_{\mathcal{I}} \\
\mathrm{d}\boldsymbol{\lambda}_{\mathcal{J}} \\
\mathrm{d}\boldsymbol{\lambda}_{\mathcal{K}} \\
\mathrm{d}\boldsymbol{\nu}
\end{bmatrix}
=
\begin{bmatrix}
-x_j \mathbf{1}^i \\
\mathbf{0} \\
\mathbf{0} \\
\mathbf{0} \\
-\nu_i \mathbf{1}^j
\end{bmatrix}
\tag{10}
$$

Since $\mathbf{x}^*$ is non-degenerate, then (10) is feasible in $\mathrm{d}\mathbf{x}$ for all $i \in \{1, \ldots, M_2\}$ and $j \in \{1, \ldots, D\}$ by same reasoning as $\frac{\partial z}{\partial \mathbf{h}}$. Applying (5) gives $\mathbf{c}^T \frac{\partial \mathbf{x}^*}{\partial G_{ij}} = \mathbf{c}^T \mathrm{d}\mathbf{x} = \boldsymbol{\lambda}^T(\mathbf{0} - \mathbf{0}\mathbf{x}) + \boldsymbol{\nu}^T(\mathbf{0} - \mathbf{1}^{ij}\mathbf{x}) = -\nu_i x_j$ where $\mathbf{1}^{ij}$ is the $M_2 \times D$ matrix with 1 for component $(i, j)$ and zeros elsewhere. The result for $\frac{\partial z}{\partial \mathbf{G}}$ then follows from $\mathbf{c}^T \frac{\partial \mathbf{x}^*}{\partial \mathbf{G}} = -\boldsymbol{\nu}^* \mathbf{x}^{*T}$ where we have slightly abused notation by dropping the leading singleton dimension of the $1 \times M_2 \times D$ Jacobian.

**Evaluating** $\frac{\partial z}{\partial \mathbf{A}}$. Consider $\frac{\partial z}{\partial A_{ij}} = -\mathbf{c}^T \frac{\partial \mathbf{x}^*}{\partial A_{ij}}$. Set $\mathrm{d}A_{ij} = 1$ and all other parameter differentials to 0. Then the right-hand side of (6) becomes

$$
\begin{bmatrix}
\mathbf{G} & \mathbf{0} & \mathbf{0} & \mathbf{0} & \mathbf{0} \\
\mathbf{D}(\boldsymbol{\lambda}_{\mathcal{I}})\mathbf{A}_{\mathcal{I}} & \mathbf{0} & \mathbf{0} & \mathbf{0} & \mathbf{0} \\
\mathbf{0} & \mathbf{0} & \mathbf{D}(\mathbf{A}_{\mathcal{J}}\mathbf{x} - \mathbf{b}_{\mathcal{J}}) & \mathbf{0} & \mathbf{0} \\
\mathbf{0} & \mathbf{0} & \mathbf{0} & \mathbf{0} & \mathbf{0} \\
\mathbf{0} & \mathbf{A}_{\mathcal{I}}^T & \mathbf{A}_{\mathcal{J}}^T & \mathbf{A}_{\mathcal{K}}^T & \mathbf{G}^T
\end{bmatrix}
\begin{bmatrix}
\mathrm{d}\mathbf{x} \\
\mathrm{d}\boldsymbol{\lambda}_{\mathcal{I}} \\
\mathrm{d}\boldsymbol{\lambda}_{\mathcal{J}} \\
\mathrm{d}\boldsymbol{\lambda}_{\mathcal{K}} \\
\mathrm{d}\boldsymbol{\nu}
\end{bmatrix}
=
\begin{bmatrix}
\mathbf{0} \\
-x_j \mathbf{1}^i \\
\mathbf{0} \\
\mathbf{0} \\
-\lambda_i \mathbf{1}^j
\end{bmatrix}
\tag{11}
$$

Since $\mathbf{x}^*$ is non-degenerate, then by similar arguments as $\frac{\partial z}{\partial \mathbf{b}}$ and $\frac{\partial z}{\partial \mathbf{G}}$ (11) is feasible in $\mathrm{d}\mathbf{x}$ for all $i \in \{1, \dots, M_1\}$ and $j \in \{1, \dots, D\}$ and the result for $\frac{\partial z}{\partial \mathbf{A}}$ follows from $\mathbf{c}^T \frac{\partial \mathbf{x}^*}{\partial \mathbf{G}} = -\boldsymbol{\lambda}^* \mathbf{x}^{*T}$. $\qquad \square$

*Proof of Corollary 1.* The result for $\frac{\partial z}{\partial \mathbf{c}}$ is direct. In linear programming, Tijssen and Sierksma [1998] showed that the existence of a non-degenerate primal solution $\mathbf{x}^*$ implies uniqueness of the dual solution $\boldsymbol{\lambda}^*, \boldsymbol{\nu}^*$ so the result for $\frac{\partial z}{\partial \mathbf{b}}$ and $\frac{\partial z}{\partial \mathbf{h}}$ follows directly. If a non-degenerate solution $\mathbf{x}^*$ is unique then matrices $\boldsymbol{\lambda}^* \mathbf{x}^{*T}$ and $\boldsymbol{\nu}^* \mathbf{x}^{*T}$ are both unique, regardless of whether $\mathbf{c} = \mathbf{0}$. In the other direction, if $\boldsymbol{\lambda}^* \mathbf{x}^{*T}$ and $\boldsymbol{\nu}^* \mathbf{x}^{*T}$ are both unique, consider two mutually exclusive and exhaustive cases: (1) when either $\boldsymbol{\lambda}^* \neq \mathbf{0}$ or $\boldsymbol{\nu}^* \neq \mathbf{0}$ this would imply $\mathbf{x}^*$ unique, and (2) when both $\boldsymbol{\lambda}^* = \mathbf{0}$ and $\boldsymbol{\nu}^* = \mathbf{0}$ in (DP) this would imply $\mathbf{c} = \mathbf{0}$, *i.e.* the primal linear program (LP) is merely a feasibility problem. The result for $\frac{\partial z}{\partial \mathbf{A}}$ and $\frac{\partial z}{\partial \mathbf{G}}$ then follows. $\qquad \square$

## Appendix D: Additional Results

Figure 7 shows the task of learning $(\mathbf{c}, \mathbf{A}, \mathbf{b})$ with a $K = 6$ dimensional parametrization $\mathbf{w}$ and 20 training observations for a $D$ dimensional decision space $\mathbf{x}$ with $M_1$ inequality constraints. The five different considered combinations of $D$ and $M_1$ are shown in the figure. The results over all problem sizes are similar to the case of $D = 10, M_1 = 80$ shown in the main paper: RS fails; COBYLA 'succeeds' on ~25% of instances; SQP succeeds on ~60–75%. As expected, instances with higher $D$, are more challenging as we observe that the success rate decreases slightly. The success curve of $\text{SQP}_{\text{bprop}}$ slightly lags those of $\text{SQP}_{\text{impl}}$ and $\text{SQP}_{\text{dir}}$ due to the overhead of backpropagating through the steps of the interior point solver. However, this computational advantage of $\text{SQP}_{\text{impl}}$ and $\text{SQP}_{\text{dir}}$ over $\text{SQP}_{\text{bprop}}$ is less obvious on LP instances with $D = 10$. For larger LP instances, the overall framework spends significantly more computation time on other components (e.g., solving the forward problem, solving (SQP)). Thus, the advantage of $\text{SQP}_{\text{impl}}$ and $\text{SQP}_{\text{dir}}$ in computing gradients is less significant in the overall performance. The $\text{SQP}_{\text{cvx}}$ implementation works better than COBYLA for most instances, but struggles to converge to the requested tolerance when more constraints are added ($M_1 = 80$, shown in Figure 3).

Figure 8 shows instances with equality constraints, where $\mathbf{G}$ and $\mathbf{h}$ also need to be learned, and the performance is similar. Note that RS failed to find a feasible $\mathbf{w}$ in all instances, caused mainly by the failure to satisfy the equality target feasibility constraints in (1a). Recall that a feasible $\mathbf{w}$ means both (1a) and (1b) are satisfied.

Figure 9 shows the performance on LPs where the dimensionality of $\mathbf{w}$ is higher. We observe that COBYLA performs poorly, $\text{SQP}_{\text{cvx}}$ makes progress but is slow, and the remaining SQP methods succeed quickly on all instances. COBYLA's poor performance is caused by the finite-difference approximation technique used in COBYLA which is inefficient in high dimension $\mathbf{w}$ space. This result demonstrates the importance of using gradient-based methods in high dimensional (in $\mathbf{w}$) NLP.

**Sensitivity of results to parameter settings** The specific results of our experiments can vary slightly with certain choices, but the larger conclusions do not change: the gradient-based SQP methods all perform similarly, and they consistently out-perform non-gradient-based methods, especially for higher-dimensional search.

Specific choices of parameter settings include the numerical tolerance used in the forward solve (e.g. $10^{-5}$ vs $10^{-8}$), algorithm termination tolerances of the COBYLA and SLSQP, and PyTorch version (v1.6 vs. nightly builds). However, we did see a degradation in "success rates" when tolerance on the forward problem was configured to be weak ($10^{-3}$), which may be caused by unstable or inaccurate gradients. The running time of the $\text{SQP}_{\text{cvx}}$ forward solver, *scs*, can be very sensitive to the numerical

Figure 7: A comparison on synthetic PLP instances as in Figure 3 but with other choices of decision variable dimension $D$ and inequality constraints $M_1$. Shown are 100 trials, where each trial includes 20 training and 20 testing instances. After training, if an instance $\mathbf{u}_i$ (of 20) for which the LP $\mathbf{c}(\mathbf{u}_i, \mathbf{w}), \mathbf{A}(\mathbf{u}_i, \mathbf{w}), \mathbf{b}(\mathbf{u}_i, \mathbf{w})$ is infeasible or unbounded, then we report a loss $\ell(\mathbf{w}) = 100$ arbitrarily and consider these to be failures. In the $M_1 = 80$ case, $\text{SQP}_{\text{cvx}}$ tends to fail for one of two reasons: its forward solver (*scs*) is slow to converge to the requested tolerance of $10^{-8}$, or *cvxpylayers* raises an exception on encountering any infeasible/unbounded instance (whereby we return $\ell = 100$); the latter behaviour is a consequence of how *cvxpylayers* handles errors, not a fundamental limitation.

Figure 8: Same as Figure 7(v) but with the additional task of learning parametric equality constraints. Specifically the experiment is configured with with decision variable dimension $D=10$, number of inequalities $M_1=80$, and number of equalities $M_2=2$.

Figure 9: A comparison on synthetic LP instances ($D=10$, $M_1=80$). Shown is the probability of achieving zero AOE training loss over time (curves), along with final loss (box plots). Each mark denotes one of 100 trials (different instances), each with one training point. Note, in this experiment we aim to learn LP coefficients directly, i.e., $\mathbf{w}$ comprises all LP coefficients, and the LP coefficients do not depend on $\mathbf{u}$. Therefore, there is only a single target $\mathbf{x}^{\text{obs}}$ for learning $\mathbf{w}$, and no testing data.

tolerance requested. For example, using the default SciPy tolerance of $10^{-8}$ and `max_iter`$=10^8$, the *scs* solver could be >100x slower than the case of using its default settings of tolerance $10^{-3}$ and `max_iter`$=2500$.

In general, the experimental results of $\text{SQP}_{\text{bprop}}, \text{SQP}_{\text{impl}}$ and $\text{SQP}_{\text{dir}}$ are largely insensitive to specific parameter settings. For example, we tried using strict tolerances and different trust region sizes for COBYLA to encourage the algorithm to search more aggressively, but these made only a small improvement to performance; these small improvements are represented in our results. We also observed that, although the homogeneous solver works slightly better when we use a strict numerical tolerance, there is no major difference in the learning results.

**Appendix E: Parametric Linear Program for Figure 6**

*Forward optimization problem for Figure 6.* The FOP formulation used is shown in (12) below.

$$
\begin{aligned}
\underset{x_1, x_2}{\text{minimize}} \quad & -w_1 u_1 x_1 - w_2 u_2 x_2 \\
\text{subject to} \quad & x_1 + x_2 \leq \max(1, u_1 + u_2) \\
& 0 \leq x_1 \leq 1 \\
& 0 \leq x_2 \leq 1
\end{aligned}
\tag{12}
$$

The two training points are generated with $\mathbf{w} = (1,1)$ at $\mathbf{u}_1 = (1, \frac{1}{3})$ and $\mathbf{u}_2 = (1, \frac{1}{3})$ with testing point $\mathbf{u}_{\text{test}} = (\frac{1}{2}, \frac{5}{6})$. PLP learning was initialized at $\mathbf{w}_{\text{ini}} = (4, 1)$ and the $\text{SQP}_{\text{impl}}$ algorithm returned $\mathbf{w}_{\text{learned}} \approx (\frac{35}{9}, \frac{4}{3})$, used to generate the learned decision map depicted in the figure.