[Reviews · NeurIPS 2020]

Review 1

Summary and Contributions: This paper presents a method for learning linear programs from data observed from optimal decisions with a strong application to inverse optimization and recovering the parameters of LPs. They provide a PyTorch implementation of an interior-point method and provide two differentiation options. --- Thanks for the rebuttal and clarifications. After the reviewing period I maintain the same view on this paper.

Strengths: Understanding how to recover LP parameters from data and inverse optimization is a timely research topic and has many potential applications in operations research and controls. Characterizing the differentiation models (backprop through the solver, implicit, and direct) is important. The ablations here are interesting and could be improved by analyzing other tradeoffs between these methods, such as compute/memory tradeoffs. The loss results in Figure 3 and Figure 5 alone make it difficult to see what differentiation method should be used in practice. The ability to handle and correct unbounded or infeasible subproblems is important and this method demonstrates the ability to handle this.

Weaknesses: Beyond the random search and COBYLA baselines, I don't have a good sense of how this method compares to related approaches in the settings they consider, for example the simpler settings of [Tan 2019].

Correctness: I see no correctness errors in this paper

Clarity: The paper is clear and well-written

Relation to Prior Work: The paper adequately describes the related work in this space

Reproducibility: Yes

Additional Feedback: One interesting use case of the unrolled derivatives is when the solver does not fully solve the problem. It would be insightful to do an experimental analysis similar to the min-differentiation ablations in [Ablin 2020, https://arxiv.org/abs/2002.03722] to see how close the derivative approximations are as the iterates converge.


Review 2

Summary and Contributions: The authors consider inverse optimization, a very important and little studied area, and propose a common-sensical approach, based on sequential quadratic programming.

Strengths: Inverse optimization, a very important and little studied area. It's natural to recast inverse linear programming as a non-linear optimisation problem (NLP), as the authors do, and it is likewise natural to solve the NLP by a sequence of convexifications (SQP).

Weaknesses: Unfortunately, the authors stop at the rather obvious suggestion that one can apply SQP to the NLP in question. The authors do not even prove the SQP converges to second-order stationary points: http://www.optimization-online.org/DB_HTML/2013/10/4102.html More generally, there is much stronger theory for SQP: https://link.springer.com/article/10.1007/s10589-014-9696-2 than the authors present for their very special case of NLP in the current paper. The implementation, while using PyTorch and many rather complicated tools, does not seem to scale beyond very small instances. It also does not acknowledge the heavy reliance on: https://github.com/cvxgrp/cvxpylayers https://web.stanford.edu/~boyd/papers/pdf/diff_cvxpy.pdf which is cited only as a pre-print, rather than the actual NeurIPS 2019 paper. Perhaps most importantly, the implementation is compared to a rather basic algorithm for generic NLP (COBYLA of Powell), rather than the state of the art in general-purpose DFO or specialised methods for inverse optimisation. It's hence largely beating the straw man.

Correctness: As far as I can tell.

Clarity: Figures 3 and 5 are rather confusing. What does "training success @ 10^-5" mean? Should the percentage be divided by 10,000?

Relation to Prior Work: No -- see weaknesses.

Reproducibility: Yes

Additional Feedback: Having read the rebuttal and the other reviews, I have amended my review and score.


Review 3

Summary and Contributions: The authors proposed to generalize the inverse optimization problem to a framework called "inverse linear optimization problem", where the coefficients of the LPs are parameterized by arbitrary functions. Then, they proposed to use the implicit-gradient-based method to differentiate through the LP to learn about the parameters.

Strengths: The authors discussed the cases when inverse optimization methods failed and compared their gradient method to the traditional inverse optimization methods. They showed that the proposed method successfully recovers the parameters of a multi-community flow problem. It is good to see comparison from inverse optimization literature in the ML settings. Also, although not explicitly mentioned in the draft, I think the authors spent considerable effort on tuning the LP solver / code to make it work under the settings.

Weaknesses: A fatal weakness of the draft is that the proposed framework is already covered by the OptNet / CVXPY layer, but there is no comparison to them. Specifically, theorem 1 is a special case of appendix B in the CVXPY paper. If the focus of the paper is on the solver, then comparison to CVXPY is necessary. From the current experiment, I cannot see the advantage over simply using the existing CVXPY layer. On the other hand, if the focus of the paper is on comparing works in inverse optimization literature, then comparisons on more recent work in IO might be necessary.

Correctness: I think the overall claims are correct.

Clarity: The paper is clearly written, and the figures are illustrative.

Relation to Prior Work: Prior works are properly discussed but not properly compared in experiments.

Reproducibility: Yes

Additional Feedback: The rebuttal addressed the concerns in my review, so I bumped the score from 5 to 6.


Review 4

Summary and Contributions: The paper contributes a gradient-based approach to 'inverse optimization' problems. Here, we observe (u, x) pairs, where x is assumed to be the optimum of an optimization problem with parameters given by a function f_w(u). The goal is to learn u. Inverse optimization is very important in a variety of applications. This paper contributes a method that allows arbitrary f_w(), such as neural networks.

Strengths: It is important to build stronger bridges between the OR and ML communities. This paper is approachable for an ML audience (it uses black-box neural networks for certain modules, it backprops through LP solvers) but also draws on sophisticated non-linear programming techniques that are not familiar to most Neurips readers. The paper is well written, is technically rigorous, and does a good job of analyzing the behavior of the methods.

Weaknesses: The paper relies on synthetic problems for evaluation. As the authors point out, this is typical of the inverse optimization literature, as it is important to have a controllable distribution over optimization problems. The experiments are technically sound. However, a real-world application, with a case study on how to formulate the various components in the objective, would be helpful.

Correctness: I am not an expert on non-linear programming, so I can't evaluate some of the details of the derivations of the method. I found the empirical methodology satisfactory.

Clarity: Yes, I found it surprisingly approachable.

Relation to Prior Work: Yes. The paper's method is applicable in more settings than prior inverse optimization work. For example, they can also learn the parameterization of the cost matrix (not just the objective weights) of an LP. Their method is more general because they do not assume a given feasible region for each optimization example.

Reproducibility: Yes

Additional Feedback: **update after author's response** I found that you addressed the other reviewers' concerns adequately.

[Author Response · NeurIPS 2020]

Dear reviewers, first we want to say that your feedback was very helpful. We see a few genuine concerns and some
misunderstandings, so we appreciate the opportunity to address them. Specific issues raised by the reviewers (R1, R2,
R3, R4) have been organized into paragraphs below. Thank you! — The Authors.

***The gradient-free baseline should be (e.g.) BARON, not COBYLA* (R2).** BARON is not applicable: a gradient-free
baseline must support either bilevel natively or must allow black-box objective and constraints (i.e., callbacks). This
also rules out CPLEX and Gurobi. IPOPT, BONMIN and Couenne are all gradient-based. YALMIP (MATLAB)
supports bilevel via reformulation as a single-level model, but imposes restrictions on how parameters appear in the
inner problem. Pyomo's `bilevel` support is restrictive and deprecated. We would like to clarify that our aim was to
compare *local* NLP methods like COBYLA and SQP, not global methods like BARON, ARGONAUT [Boukouvala
et al., 2016], or DOMINO [Beykal et al., 2020]. Of the many gradient-free solvers provided by the well-known
NLopt package, the documentation explains that only COBYLA supports arbitrary non-linear equality and inequality
constraints. (Likewise, the only such gradient-based solver in NLopt happens to be SLSQP.) Finally, note that COBYLA
was used as a state-of-the-art local NLP baseline in the ARGONAUT and and DOMINO papers.

***Compare to OptNet/CvxpyLayers* (R2, R3).** We agree that such a comparison is important for the gradient
computation component of our work. In fact, method $\text{SQP}_{\text{impl}}$ (implicit differentiation) in our experiments was
intended to be this comparison: it is our efficient implementation of OptNet specialized to LPs, and performed
well. However, as `cvxpylayers` [Agrawal et al. 2019] can implement OptNet efficiently, we agree that a compar-
ison to that implementation is informative. We used `cvxpylayers` to implement method $\text{SQP}_{\text{cvx}}$, shown at right.
As in our previous experiments, we use the same vectorized LP generation,
solve in batch mode, enable efficient re-solves, and use double precision.
We confirmed that timing differences were due to the speed of the internal
`scs` solve (fast conic solver written in C) and not any overhead. The
only difference is that the default `max_iters` for `cvxpylayers` can
terminate long before reaching the requested precision ($10^{-8}$, the SciPy
default), but we were careful to ensure that this did not limit its maximum
achievable success rate. We found `scs` can be very slow when held to the
same precision standard as the rest of our experiments (dotted line). We
in no way "relied upon" [Agrawal et al. 2019] in developing our work.

***Compare to [Tan et al. 2019]* (R1).** A comparison with their work was done but not highlighted due to two reasons:
their method is not applicable to objective error loss when learning cost with constraints, as it returns arbitrarily bad
solutions like $\mathbf{w}_1$ in Figure 1 of the paper; their forward solver implementation is much slower and occasionally fails.

***Compare to specialized inverse methods* (R2, R3).** To the best of our knowledge, there are no other inverse
optimization (IO) methods in the literature that can be directly applied to the problem instances we investigated. Most
IO methods make strong assumptions or address one aspect of the problem. Our approach is qualitatively more flexible.

***Scalability concerns* (R2).** We are actually very excited about the scalability opportunities of our approach! First,
consider that the forward solver can be any specialized algorithm that returns primal and dual variables for the problem
at hand, not just a general-purpose LP solver; this means the forward solve takes a much smaller fraction of compute.
Second, with fast forward solvers the speed advantage of our closed form gradient becomes important, because solving
a large system (OptNet/CvxpyLayers-style) scales much worse than simply computing an outer product (our approach);
even on these modest instances, our closed form gradients are >100x faster than `CvxpyLayer.backward`. Other
well-known IO approaches, such as single-level models [Keshavarz et al., 2011], cannot exploit fast solvers for the
forward problem (i.e., the inner problem of our formulation) and must rely on general-purpose machinery like CPLEX.

***Isn't Theorem 1 a special case of Appendix B in [Agrawal et al., 2019]?* (R3).** No. Although LPs are a special case
of convex programs, our theorem applies only to objective error loss (not considered by [Amos & Kolter, 2017] nor
[Agrawal et al., 2019]) and cannot be derived by merely specializing the steps of their existing proofs.

***Prove SQP convergence to second-order stationary points* (R2).** We agree that such a proof would be informative,
but characterizing the theoretical properties of a particular gradient-based NLP method (such as SQP) in this setting
was not our goal. Different parametrizations may or may not satisfy the technical conditions required (e.g., to guarantee
that $\mathbf{B}$ matrix remains a good Hessian estimate), but such analysis was not the focus of our work.

***The paper just applies SQP to an NLP, which is obvious* (R2).** The IO formulation itself is our contribution and,
despite seeming "natural" (we agree!), has not been proposed before in either the `cvxopt`/OptNet body of work nor
in the IO-specific literature. So, the paper is not just about "applying SQP to an NLP." Our framework provides the
most general approach for inverse LPs to date. Further, due to the importance of LPs and the existence of many fast
algorithms for specialized LPs, we believe that our fast gradient formula has long-term value.

[Meta-Review · NeurIPS 2020]

All the reviewers agreed that this paper studies an interesting problem and a novel method is presented. Although some reviewers initially raised a concern regarding the novelty, the authors provided a clear response and the concern was appropriately addressed. All reviewers and I agreed to suggest acceptance of this submission. Note however that several reviewers pointed out some important concerns. Please consider revising your paper to address them before submitting camera-ready.